# IMAGE AND VIDEO TOKENIZATION WITH BINARY SPHERICAL QUANTIZATION

**Yue Zhao**
UT Austin
yzhao@cs.utexas.edu

**Yuanjun Xiong** *
MThreads AI
bitxiong@gmail.com

**Philipp Krähenbühl**
UT Austin
philkr@cs.utexas.edu

## ABSTRACT

We propose a new transformer-based image and video tokenizer with Binary Spherical Quantization (BSQ). BSQ projects the high-dimensional visual embedding to a lower-dimensional hypersphere and then applies binary quantization. BSQ is (1) parameter-efficient without an explicit codebook, (2) scalable to arbitrary token dimensions, and (3) compact: compressing visual data by up to $100\times$ with minimal distortion. Our tokenizer uses a transformer encoder and decoder with simple block-wise causal masking to support variable-length videos as input. The resulting BSQ-ViT achieves state-of-the-art visual reconstruction quality on image and video reconstruction benchmarks with $2.4\times$ throughput compared to the best prior methods. Furthermore, by learning an autoregressive prior for adaptive arithmetic coding, BSQ-ViT achieves comparable visual compression results with commonly used compression standards, *e.g.* JPEG2000/WebP for images and H.264/H.265 for videos. BSQ-ViT also enables masked language models to achieve competitive image synthesis quality to GAN and diffusion approaches.

## 1 INTRODUCTION

Learned discrete image and video tokenization allows for state-of-the-art visual compression (Daede et al., 2016; Agustsson et al., 2017; El-Nouby et al., 2023), recognition (Yu et al., 2022; Bao et al., 2022; Zhou et al., 2022; Wang et al., 2022) and generation (Van Den Oord et al., 2017; Esser et al., 2021; Chang et al., 2022). These models follow a proven recipe from large language modeling (Brown et al., 2020; Achiam et al., 2023; Touvron et al., 2023): Tokenize input and outputs into discrete units and learn an auto-regressive model to predict this tokenized stream one token at a time. The most widely used approach for image encoding is Vector-Quantized Variational Auto-Encoder (VQ-VAE) (Van Den Oord et al., 2017). They encode inputs in continuous latent embeddings and map them to a learned codebook through nearest-neighbor lookup. However, VQ-VAE style approaches have two drawbacks: First, most image encoders are built upon convolutional networks (CNN) (Esser et al., 2021; Podell et al., 2023). Adapting spatial convolution for images to spatial-temporal convolution for videos requires non-trivial architectural changes (Ge et al., 2022; Yu et al., 2023; 2024) with increased computational cost. Treating videos as a sequence of images leads to a suboptimal quantization (Yu et al., 2023). Second, vector quantization (VQ) scales poorly with the codebook size. The runtime scales linearly with the codebook size, and the codebook easily overfits on smaller datasets (Yu et al., 2024). This is especially troubling for video inputs, as they rely on larger codebooks to represent both static visual patterns and dynamic motion patterns.

This paper proposes a unified visual tokenizer based on a Vision Transformer and Binary Spherical Quantization (BSQ). The Transformer-based encoder-decoder leverages a block-wise causal mask and uses only visual tokens from the current or past timestamps for reconstruction (Figure 3). BSQ first projects the high-dimensional visual embedding of the encoder to a lower-dimensional hypersphere and then applies binary quantization. The transformer encoder, decoder, and BSQ are seamlessly integrated into the VQ-GAN (Esser et al., 2021) framework and trained end-to-end.

Our proposed visual tokenizer features several advantages. First, the Transformer-based encoder-decoder shows a Pareto improvement in visual reconstruction quality and computational efficiency

---

*Now at Adobe Firefly

compared to standard CNNs. Second, the block-wise causal design unifies images and videos as input at training and supports variable-length videos at inference. BSQ constructs an implicit codebook whose effective vocabulary grows exponentially with the spherical dimension with no learned parameters. The increasing codebook size consistently yields better reconstruction results. Compared to Lookup-free Quantization (LFQ) (Yu et al., 2024), a recent technique that also builds an implicit codebook based on scalar quantization (SQ), BSQ has a bounded quantization error and is easier to train. Furthermore, we show that the soft quantization probability in BSQ reduces to a simple product of multiple channel-independent Bernoulli distributions, leading to efficient entropy regularization during training. Specifically, we show how a factorized approximation to the entropy for soft quantization of $L$ bits reduces the theoretical computation complexity from $O(2^L \times L)$ to $O(L)$ with minimal approximation error, and negligible performance degradation in practice.

We validate the effectiveness of BSQ-ViT on visual reconstruction and compression benchmarks. On image reconstruction, our model archives a state-of-the-art visual reconstruction quality by both pixel-level and semantic metrics. In particular, our best-performing BSQ-ViT achieves a reconstruction FID of 0.41 on ImageNet-1k val, a 43% reduction compared to the runner-up (SDXL-VAE (Podell et al., 2023)), while being 2.4× faster. On video reconstruction, our best model reduces FVD on UCF-101 by more than half (8.62 → 4.10). By further learning an autoregressive prior for adaptive arithmetic coding, BSQ-ViT achieves comparable results on video compression with conventional compression standards, *e.g.* H.264 and HEVC. By learning a masked language model, BSQ-ViT enables image generation with similar quality to BigGAN (Brock et al., 2018) and ADM (Dhariwal & Nichol, 2021). Code is available at https://github.com/zhaoyue-zephyrus/bsq-vit.

## 2 RELATED WORK

**Visual Tokenization.** VQ-VAE (Van Den Oord et al., 2017) introduced the concept of discrete tokenized bottlenecks in auto-encoder architectures. Recent improvements include better training objectives (Ramesh et al., 2021; Esser et al., 2021), increasing VQ codebook usage (Yu et al., 2022; Zheng & Vedaldi, 2023), replacing VQ with product quantization (PQ) (El-Nouby et al., 2023) or scalar quantization (SQ) (Mentzer et al., 2024), and employing stronger generative models (Esser et al., 2021; Chang et al., 2022). Image tokenizers are trivially extended to video by tokenizing individual frames (Blattmann et al., 2023). However, this ignores dynamic motions and leads to suboptimal tokenization: The same visual information is compressed repeatedly across frames.

**Video Tokenization.** Dedicated video tokenizers make better use of temporal correlations in the input signal. Yan et al. (2021) proposes 3D (de-)convolutions in VQ-VAE for video generation. TATS (Ge et al., 2022) replaces zero padding with replicate padding to mitigate the temporal corruption when video length varies. Yu et al. (2023) introduce central inflation of pretrained 2D convolutional filters to 3D and further make them causal (Yu et al., 2024). Phenaki (Villegas et al., 2022) adopts a factorized causal video vision Transformer (Arnab et al., 2021) (C-ViViT), which improves efficiency but sacrifices modeling complex motion across time. All these methods focus on generation while we demonstrate that a good video tokenizer can perform well in compression.

**Neural Compression.** Since Shannon established the fundamental source coding theorem (Shannon, 1948), it has formed the basis of lossless compression (Huffman, 1952; Pasco, 1976; Rissanen & Langdon, 1979; Duda, 2009) with probabilistic models including RNN (Mikolov, 2012; Goyal et al., 2019), CNN (Van den Oord et al., 2016; Van Den Oord et al., 2017), VAE (Townsend et al., 2019; 2020), and Transformers (Bellard, 2019; Delétang et al., 2024). Mentzer et al. (2019) presents a fast hierarchical probabilistic model (L3C) for lossless image compression. Delétang et al. (2024) show that LLMs trained primarily on text, *e.g.* Llama 2 (Touvron et al., 2023) and Chinchilla (Hoffmann et al., 2022), are general-purpose compressors for text, images, and audio. However, these LLMs are too big and slow to make this compression practical. Our tokenizer presents a lighter-weight alternative: Tokenization performs initial local lossy compression, while a lightweight and thus computationally efficient sequence model (∼300M) compresses the global video structure.

**Video compression.** Most high-performing modern video compression methods rely on hybrid coders that combine transform coding (Goyal, 2001; Ballé et al., 2020) and motion compensation (Wiegand et al., 2003; Sullivan et al., 2012). Such belief continues in most of the recently popularized learning-based solutions (Lu et al., 2019; Rippel et al., 2019; Agustsson et al., 2020; Li et al., 2021). VCT (Mentzer et al., 2022) proposes a Transformer-based temporal entropy model to learn motion implicitly. However, VCT requires a heavily-engineered image compression model (He

et al., 2022) and has a short temporal context window. In this work, we show that a learned video tokenizer combined with an arithmetic coder modeled by a sequence model achieves competitive compression results without explicitly modeling motion.

## 3 PRELIMINARIES

A tokenization-based compression algorithm has three basic steps: A visual tokenizer, *i.e.* VQ-VAE or LFQ, translates raw visual inputs to a discrete set of tokens. A sequence model then predicts an auto-regressive probability distribution over these discrete tokens. Finally, arithmetic coding converts this distribution into a compressed representation.

**Visual Tokenization.** VQ-VAE (Van Den Oord et al., 2017) introduced the concept of learning discrete visual representation with an auto-encoder architecture and a bottleneck module in between with vector quantization (VQ). Given a video $\boldsymbol{X} \in \mathbb{R}^{T \times H \times W \times 3}$, an encoder $\mathcal{E}$ produces a set of $d$-dimensional latent embeddings $\boldsymbol{Z} = \mathcal{E}(\boldsymbol{X}) \in \mathbb{R}^{\left(\frac{T}{r} \times \frac{H}{p} \times \frac{W}{p}\right) \times d}$ with a spatial-temporal downsample factor of $r \times p \times p$. The bottleneck module $q$ then transforms the real-valued latent embeddings into some discrete tokens $\hat{\boldsymbol{z}} = q(\boldsymbol{z})$. In VQ, the quantizer $q_{VQ}$ assigns each $\boldsymbol{z} \in \boldsymbol{Z}$ to the closest entry in a learnable code in a codebook $\boldsymbol{C} = [\boldsymbol{c}_1 \cdots \boldsymbol{c}_K] \in \mathbb{R}^{K \times d}$

$$\hat{\boldsymbol{z}} = q_{VQ}(\boldsymbol{z}) = \boldsymbol{c}_k = \arg\min_{\boldsymbol{c}_{\hat{k}} \in \boldsymbol{C}} \|\boldsymbol{z} - \boldsymbol{c}_{\hat{k}}\|_2. \tag{1}$$

Here, $K$ is the vocabulary size of the codebook and the integer $k$ is the discretized token representation of $\boldsymbol{z}$ which can be stored in $\lceil \log(K) \rceil$ bits. A decoder $\mathcal{G}$ maps the discretized tokens back into a visual representation $\hat{\boldsymbol{X}} = \mathcal{G}(\hat{\boldsymbol{Z}})$. The entire network ($\mathcal{E}$, $\mathcal{G}$, and $q$) is end-to-end trainable and minimizes an MSE loss $\mathcal{L}_{\text{MSE}} = \|\hat{\boldsymbol{X}} - \boldsymbol{X}\|_2$ using STE (Bengio et al., 2013) to propagate gradients through the quantization bottleneck. More recent quantizers rely on a perceptual $\mathcal{L}_{\text{LPIPS}}$ and adversarial $\mathcal{L}_{\text{GAN}}$ loss for better visual quality (Zhang et al., 2018; Esser et al., 2021),

$$\underset{\mathcal{E},\mathcal{G},q}{\text{minimize}}\, \mathbb{E}_{\boldsymbol{X}} \left[ \mathcal{L}_{\text{VQ}}(\mathcal{E},\mathcal{G},q) + \eta \mathcal{L}_{\text{LPIPS}}(\mathcal{E},\mathcal{G},q) + \lambda \mathcal{L}_{\text{GAN}}(\mathcal{E},\mathcal{G},q) \right], \tag{2}$$

where the quantization loss term $\mathcal{L}_{\text{VQ}}$ emulates online clustering to learn $\boldsymbol{c}_k$. The main issue with VQ-VAE is that Vector Quantization scales poorly with increasing vocabulary size $K$ (Yu et al., 2024). Remedies include using a smaller code dimension (Yu et al., 2022), introducing stochasticity (Takida et al., 2022), reviving "dead" codevectors (Zheng & Vedaldi, 2023), and regularizing with a commitment loss (Van Den Oord et al., 2017):

$$\mathcal{L}_{\text{commit}}(\hat{\boldsymbol{z}}, \boldsymbol{z}) = \|\operatorname{sg}(\hat{\boldsymbol{z}}) - \boldsymbol{z}\|, \tag{3}$$

where $\operatorname{sg}(\cdot)$ denotes the stop-gradient operation.

**Lookup-Free Quantization** (LFQ) (Yu et al., 2024) uses a fixed implicit codebook $\boldsymbol{C}_{LFQ} = \{-1, 1\}^L$ as corners of a hypercube in $L$ dimensional space. The best vector quantizer for this implicit codebook is the binary quantization $q_{LFQ}(\boldsymbol{z}) = \operatorname{sign}(\boldsymbol{z})$. To optimize for an effective latent code and encourage code usage, they use an additional entropy objective (Jansen et al., 2020):

$$\mathcal{L}_{\text{entropy}} = \mathbb{E}\left[H(q(\boldsymbol{z}))\right] - \gamma H\left[\mathbb{E}\left[q(\boldsymbol{z})\right]\right], \tag{4}$$

where both entropy terms rely on a soft quantization (Agustsson et al., 2017)

$$\hat{q}(\boldsymbol{c}|\boldsymbol{z}) = \frac{\exp(-\tau(\boldsymbol{c} - \boldsymbol{z})^2)}{\sum_{\boldsymbol{c} \in \boldsymbol{C}_{LFQ}} \exp(-\tau(\boldsymbol{c} - \boldsymbol{z})^2)} \tag{5}$$

to guarantee the loss is differentiable. The final loss $\mathcal{L}_{LFQ}$ is a combination of $\mathcal{L}_{\text{MSE}}$, $\mathcal{L}_{\text{commit}}$, $\mathcal{L}_{\text{LPIPS}}$, $\mathcal{L}_{\text{GAN}}$, and $\mathcal{L}_{\text{entropy}}$. The main computational bottleneck in LFQ is the entropy optimization of a higher-dimensional codebook, as it involves summation over $2^L$ implicit codebook entries.

Both VQ-VAE and LFQ lossily compress visual inputs into $N$ discrete tokens $[k_1, \ldots, k_N]$, where $k_i \in \{1, \ldots K\}$, in $N \lceil \log K \rceil$ bits. Neither tokenization strategy exploits the global image or video structure well. A sequence model with lossless arithmetic coding better fits this global structure.

**Arithmetic Coding** (AC) (Pasco, 1976; Rissanen & Langdon, 1979; Witten et al., 1987) offers a way of constructing a bitstream with near-optimal length by leveraging the statistical property of the coding distribution. Given a distribution over token streams $P_t : \{1, \cdots, K\}^n \mapsto (0, 1]$, arithmetic

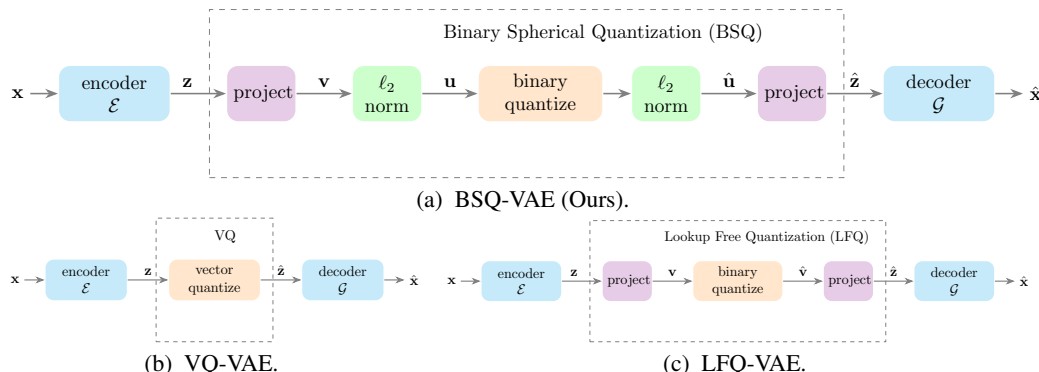

Figure 1: **Variational Auto-Encoders (VAE) with different bottlenecks (BSQ, VQ, and LFQ).**

coding looks to encode the token stream in $(-\lceil \log P_t(k_1, \ldots, k_N) \rceil + 1)$ bits. The most common token distribution is an auto-regressive model

$$P_t(k_1, \ldots, k_N) = P_t(k_1)P_t(k_2|k_1) \ldots P_t(k_N|k_1, \ldots, k_{N-1}) \tag{6}$$

for which efficient incremental encoding and decoding algorithms exist (Mentzer et al., 2022).

## 4 TRANSFORMER-BASED VISUAL TOKENIZER WITH BSQ

Our video tokenizer follows an encoder-decoder architecture with a discretization bottleneck as illustrated in Figure 1a. It combines a transformer-based encoder, a transformer-based decoder, and a Binary Spherical Quantization (BSQ) layer. BSQ projects the latent code into a lower-dimensional spherical space, applies binary quantization, and then projects the result back up into the decoder's latent space. This projection onto a low-dimensional spherical space has several theoretical advantages: The approximation error of the quantizer is bounded and much of the entropy computation factorizes along individual dimensions. These advantages result in experimental improvements as well. BSQ converges quicker and to a better tokenizer than other quantization schemes.

### 4.1 BINARY SPHERICAL QUANTIZATION

Binary Spherical Quantization (BSQ) optimizes over an implicit codebook $C_{BSQ} = \{-\frac{1}{\sqrt{L}}, \frac{1}{\sqrt{L}}\}^L$, a hypercube projected onto a unit sphere. Each corner $c_k \in C_{BSQ}$ of a hypercube corresponds to a unique token $k$. The quantizer works as follows: it projects some high-dimensional latent embedding $z$ to a lower-dimensional unit hypersphere $u$, applies binary quantization per axis $\hat{u} = \text{sign}(u)$, and back-projects to the quantized vector in the original latent space $\hat{x}$, as shown in Figure 1a. Specifically, we start with an encoded visual input $z = \mathcal{E}(x) \in \mathbb{R}^d$. We first linearly project the latent embedding to $L$ dimensions $v = \text{Linear}(z) \in \mathbb{R}^L$, where $L \ll d$. Next, we project $v$ onto the unit sphere $u = \frac{v}{|v|}$, and perform binary quantization to each dimension of $u$ independently $\hat{u} = \frac{1}{\sqrt{L}} \text{sign}(u)$, where $\text{sign}(x)$ is the sign function. To keep outputs on the unit sphere, we map $\text{sign}(0) \to 1$. We use a Straight-Through Estimator (STE) (Bengio et al., 2013) to make the operator differentiable, $\text{sign}_{\text{STE}}(x) = \text{sg}(\text{sign}(x) - x) + x$, where $\text{sg}(\cdot)$ denotes the stop-gradient operation. Finally, we back-project the quantized $\hat{u}$ to the $d$-dimensional space $\hat{z} = \text{Linear}(\hat{u}) \in \mathbb{R}^d$.

BSQ has a few appealing properties: As with LFQ, the implicit codebook entry is parameter-free and easy to compute. Unlike LFQ, a soft quantization of BSQ has a simple probabilistic interpretation, which leads to efficient entropy computation in an entropy loss $\mathcal{L}_{\text{entropy}}$. Finally, BSQ's quantization error is bounded, which empirically leads to much faster and better convergence than LFQ.

**Efficient implicit code assignment.** At inference time, we map a projected embedding $v$ to a token through simply binarization $k = \sum_{i=1}^{L} 1_{[v_i > 0]} 2^{i-1}$, where $1_{[\cdot]}$ is the indicator function. The inverse mapping uses the bitshift and the bitwise AND operations.

**Soft BSQ and entropy.** To best use the entire range of the implicit codebook $C_{BSQ}$, we use the entropy loss $\mathcal{L}_{\text{entropy}} = \mathbb{E}_u[H(q(u))] - \gamma H[\mathbb{E}_u[q(u)]]$ (Jansen et al., 2020). To compute this entropy loss we first derive a soft quantization scheme (Agustsson et al., 2017). Since both codebook

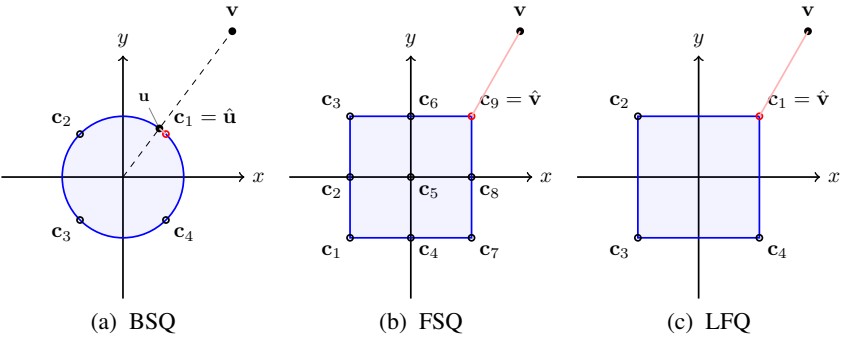

(a) BSQ        (b) FSQ        (c) LFQ

Figure 2: **Illustration of BSQ compared to LFQ and FSQ in the simplest case ($L = 2$).** In FSQ, we consider each channel to have 3 possible values $\{\pm 1, 0\}$. The Voronoi diagram for both FSQ and LFQ looks like hypercubes partitioning the entire space while BSQ's looks like a hypersphere evenly divided by $2^L$ centroids.

entries and inputs to the quantizer are unit vectors, the soft quantization is a distribution

$$\hat{q}(\boldsymbol{c}|\boldsymbol{u}) = \frac{\exp(\tau \boldsymbol{c}^\top \boldsymbol{u})}{\sum_{\boldsymbol{c} \in \boldsymbol{C}_{BSQ}} \exp(\tau \boldsymbol{c}^\top \boldsymbol{u})} = \prod_{d=1}^{L} \sigma\left(2\tau c_d u_d\right), \tag{7}$$

where $\sigma$ is a sigmoid function, and the overall soft quantizer is independent along each dimension. See Section C.1 for a derivation. This allows for an efficient computation of the first entropy term

$$\mathbb{E}_{\boldsymbol{u}}\left[H(\hat{q}(\boldsymbol{c}|\boldsymbol{u}))\right] = \mathbb{E}_{\boldsymbol{u}}\left[\sum_{d=1}^{L} H(\hat{q}(c_d|u_d))\right]. \tag{8}$$

See Section C.2 for a derivation. Instead of reasoning over distributions over the entire codebook, which is exponentially large, we instead treat each dimension independently. The resulting entropy computation is linear to the dimension $L$ of the bottleneck.

Unfortunately, the second entropy term cannot directly use the same independence assumption, as dimensions in the expected value $\mathbb{E}_{\boldsymbol{u}}[\hat{q}(\boldsymbol{c}|\boldsymbol{u})]$ are correlated through the distribution of $\boldsymbol{u}$. We find the closest factorized distribution $\tilde{q}(\boldsymbol{c}) = \prod_{d=1}^{K} \tilde{q}(c_d)$ to $\mathbb{E}_{\boldsymbol{u}}[\hat{q}(\boldsymbol{c}|\boldsymbol{u})]$, and instead minimize the entropy of the approximate distribution. As we will show in Section C.3 the best approximation in terms of the KL-divergence $\tilde{q}(c_d) = \mathbb{E}_{\boldsymbol{u}_d}[\hat{q}(c_d|u_d)]$. The final approximate entropy term to maximize is

$$H(\mathbb{E}_{\boldsymbol{u}}\left[\hat{q}(\boldsymbol{c}|\boldsymbol{u})\right]) \approx H(\tilde{q}(\boldsymbol{c})) = \sum_{d=1}^{L} H(\mathbb{E}_{\boldsymbol{u}_d}[\hat{q}(c_d|u_d)]). \tag{9}$$

As we will show in Section C.3 this approximation is an upper bound to the true entropy, but empirically closely tracks the true entropy. This entropy term is again efficient for evaluation.

**Quantization error in BSQ.** Most quantizers use straight-through gradient estimates during training (Yu et al., 2024; Van Den Oord et al., 2017). Though simple to implement, it assumes that the gradients for an unquantized $\boldsymbol{u}$ and quantized $\hat{\boldsymbol{u}}$ bottleneck are almost the same, which only holds if the quantization error $d(\boldsymbol{u}, \hat{\boldsymbol{u}}) = \|\boldsymbol{u} - \hat{\boldsymbol{u}}\|$ is small. As we show in Section C.4, this is true for BSQ:

$$\mathbb{E}_{\boldsymbol{u}}\left[d(\boldsymbol{u}, \hat{\boldsymbol{u}})\right] < \sqrt{2 - 2/\sqrt{L}} < \sqrt{2}. \tag{10}$$

**Relation to other quantization methods.** BSQ is closely connected to many concepts introduced in information and coding theories. LFQ (Yu et al., 2024) uses the same binarization technique as BSQ but does not normalize its output. This leads to an unbounded quantization error and does not allow for as simple of a soft quantization for entropy computation. A pictural comparison between LFQ and BSQ is shown in Figure 2 and a summary is provided in Table 7. Spherical Vector Quantization (SVQ) (Hamkins & Zeger, 2002) also ensures all code vectors have a pre-defined radius. However, SVQ assumes a variety of radii, which have to be encoded by an additional gain quantizer. In our case, the source code is the output of a learned encoder $\mathcal{E}$. Therefore, the unit radius assumption is sound, and the gain quantizer can be avoided. Pyramid Vector Quantization (PVQ) (Fischer, 1986) assumes all code vectors have a constant $\ell_1$ norm, but the $\ell_1$ normalized centroids partition the hypersphere less uniformly than $\ell_2$.

## 4.2 TOKENIZATION NETWORK WITH CAUSAL VIDEO TRANSFORMER

We propose to use Vision Transformer (ViT) (Dosovitskiy et al., 2021) to model both the encoder and decoder due to its better computational efficiency and higher reconstruction quality.

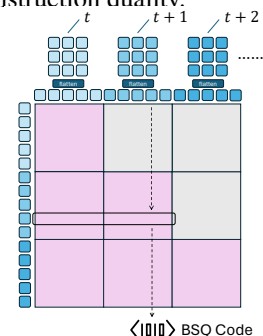

**Video Transformer.** We start from ViT-VQGAN (Yu et al., 2022) and extend it to take videos as input. We divide an input video $\boldsymbol{X} \in \mathbb{R}^{T \times H \times W \times 3}$ into non-overlapping patches of size $1 \times p \times p$. These patches are flattened into a 1D sequence, linearly projected, and passed through a stack of $N$ Transformer Encoder layers to yield the latent representation, $(\boldsymbol{z}_1, \cdots, \boldsymbol{z}_N)$. The decoder takes the same architecture, maps the latent embeddings $\hat{\boldsymbol{z}}$ back to the pixel space, and regroups them into the original shape. $(\hat{\boldsymbol{x}}_1, \cdots, \hat{\boldsymbol{x}}_N) = \text{MLP}(\text{TransformerDecoder}(\hat{\boldsymbol{z}}_1, \cdots, \hat{\boldsymbol{z}}_N))$, where MLP is a decoding head with a two-layer MLP, *i.e.* Linear ∘ Tanh ∘ Linear.

**Blockwise Causal Attention.** During training, we always assume the input video has $T$ frames, which might not hold at inference. Padding shorter video segments to $T$ frames works but wastes a lot of bits, especially in the context of compression. To handle variable-length videos, we propose a simple blockwise causal masked attention analogous to causal attention in language modeling (Vaswani et al., 2017). It specifies that only those tokens at time $t$ or earlier are used to reconstruct the visual tokens at time $t$.

Figure 3: Given an input video, block-wise causal masked attention enables the Transformer encoder to only use patches from current or past timestamps to encode each visual patch and later translate it into a BSQ code.

$$(\boldsymbol{z}_{(t-1) \times \frac{H}{p} \times \frac{W}{p} + 1}, \cdots, \boldsymbol{z}_{t \times \frac{H}{p} \times \frac{W}{p}}) = \text{TransformerEncoder}\left(\boldsymbol{x}_1, \cdots, \boldsymbol{x}_{t \times \frac{H}{p} \times \frac{W}{p}}\right),$$

$$(\hat{\boldsymbol{z}}_{(t-1) \times \frac{H}{p} \times \frac{W}{p} + 1}, \cdots, \hat{\boldsymbol{z}}_{t \times \frac{H}{p} \times \frac{W}{p}}) = q_{LFQ}\left(\boldsymbol{z}_{(t-1) \times \frac{H}{p} \times \frac{W}{p} + 1}, \cdots, \boldsymbol{z}_{t \times \frac{H}{p} \times \frac{W}{p}}\right),$$

$$(\hat{\boldsymbol{x}}_{(t-1) \times \frac{H}{p} \times \frac{W}{p} + 1}, \cdots, \hat{\boldsymbol{x}}_{t \times \frac{H}{p} \times \frac{W}{p}}) = \text{MLP}\left(\text{TransformerDecoder}\left(\hat{\boldsymbol{z}}_1, \cdots, \hat{\boldsymbol{z}}_{t \times \frac{H}{p} \times \frac{W}{p}}\right)\right).$$

This can be efficiently implemented with a blockwise causal attention mask written in a blockwise lower triangle matrix in Figure 3. When $T = 1$, the proposed encoder-decoder reduces to a ViT with a full attention mask. Therefore, we can easily train it using a mixture of images and videos.

We use factorized spatial-temporal position embedding to encode the temporal information. Specifically, we add a set of zero-initialized temporal position embeddings $\text{PE}_t \in \mathbb{R}^{T \times d}$ to the original spatial ones $\text{PE}_s \in \mathbb{R}^{N \times d}$ in the image tokenizer, *i.e.* $\text{PE}[i, :, :] = \text{PE}_t[i, \text{None}, :] + \text{PE}_s[\text{None}, :, :]$.

**Training the Video Tokenizer from an Image Tokenizer.** For training efficiency, we first train an image tokenizer on image data and then fine-tune it to be a video tokenizer. Though previous works (Wang et al., 2022; Blattmann et al., 2023) argue that a pre-trained image tokenizer can be used for videos as is, we observe that the video tokenizer after fine-tuning demonstrates much higher reconstruction quality on video benchmarks, see Section 5.1. The gain is further magnified when the effective vocabulary size becomes larger. We hypothesize that such increased vocabulary size, enabled by the proposed BSQ, is handy for learning video-specific motion and blur. In contrast, vanilla VQ methods fail to maintain high codebook usage when the codebook size exceeds 16K.

**Optimizing the Visual Tokenizer.** Following VQGAN (Esser et al., 2021), we use a perceptual loss (Zhang et al., 2018) and an adversarial loss (Goodfellow et al., 2014). We use StyleGAN (Karras et al., 2019) as the discriminator since ViT-VQGAN (Yu et al., 2022) reports it is much easier to train than PatchGAN (Isola et al., 2017). When we fine-tuned the tokenizer on videos, unlike MAGVIT or TATS, we did not inflate StyleGAN to be a 3D discriminator. Instead, we pass all reconstructed frames individually to the vanilla StyleGAN and sum up the losses.

## 5 EXPERIMENTS

We train the image tokenization model on the training set of ImageNet ILSVRC2012 (Russakovsky et al., 2015) and evaluate the image reconstruction result on the validation set of MS-COCO (Lin et al., 2014) and ImageNet, denoted by **COCO 2017val** and **ImageNet-1k** respectively. We fine-tune the video tokenization model on **UCF-101** (Soomro et al., 2012) and conduct video compression experiments on two standard benchmarks, *i.e.* MCL-JCV (Wang et al., 2016) and UVG (Mercat et al., 2020). We leave dataset statistics and implementation details in Section E.

**Evaluation metrics.** For tokenization, we report perceptual metric (LPIPS-AlexNet) (Zhang et al., 2018), PSNR, SSIM (Wang et al., 2004), and Fréchet Inception/Video Distance (FID/FVD) (Heusel

Table 1: **Image reconstruction results on COCO2017 and ImageNet-1K** ($256 \times 256$). The "data" column refers to the training data: CC for CC3M, YF for YFCC100M, OImg for OpenImages, LAION for LAION-5B, IN for ImageNet, and "?" for unknown source. The "arch." column shows the encoder/decoder architecture: C for ConvNets with Self-Attention, and T-B for ViT-Base. The "# bits" column refers to the effective number of bits per token defined in Section 5.1. $^{\#}$ is obtained by multiplying the latent dimension with the precision. The "TP" column means the inference throughput (images/second) per GPU. $^{\dagger}$The number is taken from the paper. Note that STDs of PSNR, SSIM, and LPIPS are computed across samples instead of multiple runs.

| Method | Data | Arch. | Quant. | Param. | # bits | TP↑ | COCO2017 val | | | | ImageNet-1k val | | | |
| --- | --- | --- | --- | --- | --- | --- | --- | --- | --- | --- | --- | --- | --- | --- |
| | | | | | | | PSNR↑ | SSIM↑ | LPIPS↓ | rFID↓ | PSNR↑ | SSIM↑ | LPIPS↓ | rFID↓ |
| DALL-E dVAE (Ramesh et al., 2021) | CC+YF | C | VQ | 98M | 13 | 34.0 | 25.15 ±3.49 | .7497 ±.1124 | .3014 ±.1221 | 55.07 | 25.46 ±3.93 | .7385 ±.1343 | .3127 ±.1480 | 36.84 |
| MaskGIT (Chang et al., 2022) | IN-1k | C | VQ | 54M | 10 | 37.6 | 17.52 ±2.75 | .4194 ±.1619 | .2057 ±.0473 | 8.90 | 17.93 ±2.93 | .4223 ±.1827 | .2018 ±.0543 | 2.23 |
| ViT-VQGAN (Yu et al., 2022) | IN-1k | T-B | VQ | 182M | 13 | $^{\dagger}$7.5 | - | - | - | - | - | - | - | $^{\dagger}$1.55 |
| SD-VAE 1.x (Rombach et al., 2022) | OImg | C | VQ | 68M | 10 | 22.4 | 21.78 ±3.41 | .6139 ±.1430 | .1042 ±.0345 | 6.79 | 22.12 ±3.79 | .6046 ±.1663 | .1039 ±.0409 | 1.52 |
| SD-VAE 1.x (Rombach et al., 2022) | OImg | C | VQ | 68M | 14 | 22.4 | 22.54 ±3.55 | .6470 ±.1409 | .0905 ±.0323 | 6.07 | 22.82 ±3.97 | .6354 ±.1644 | .0912 ±.0390 | 1.23 |
| SD-VAE 1.x (Rombach et al., 2022) | OImg | C | KL | 68M | $^{\#}$64 | 22.4 | 21.68 ±3.32 | .6375 ±.1375 | .0985 ±.0309 | 5.94 | 21.99 ±3.74 | .6275 ±.1600 | .0980 ±.0371 | 1.35 |
| SD-VAE 2.x (Podell et al., 2023) | OImg+ LAION | C | KL | 84M | $^{\#}$64 | 18.9 | 24.82 ±3.64 | .7202 ±.1241 | .0694 ±.0344 | 4.63 | 25.08 ±4.11 | .7054 ±.1469 | .0731 ±.0448 | 0.78 |
| SDXL-VAE (Podell et al., 2023) | OImg+ LAION+? | C | KL | 84M | $^{\#}$64 | 18.9 | 25.11 ±3.91 | .7433 ±.1240 | .0623 ±.0289 | 4.23 | 25.38 ±4.41 | .7276 ±.1469 | .0666 ±.0373 | 0.72 |
| Ours | IN-1k | T-B | BSQ | 174M | 18 | **45.1** | 25.08 ±3.57 | .7662 ±.0993 | .0744 ±.0295 | 5.81 | 25.36 ±4.02 | .7578 ±.1163 | .0761 ±.0358 | 1.14 |
| Ours | IN-1k | T-B | BSQ | 174M | 36 | **45.1** | 27.64 ±3.74 | .8485 ±.0704 | .0412 ±.0199 | 3.42 | 27.88 ±4.26 | .8410 ±.0821 | .0432 ±.0253 | **0.41** |
| Ours (w/. EMA) | IN-1k | T-B | BSQ | 174M | 36 | **45.1** | **27.92** ±3.78 | **.8526** ±.0698 | **.0380** ±.0187 | 3.34 | **28.14** ±4.32 | **.8414** ±.0814 | **.0400** ±.0237 | 0.45 |

et al., 2017; Unterthiner et al., 2019). To distinguish it from generation, we denote it as rFID/rFVD. For generation, we report FID, Inception Score (IS) (Salimans et al., 2016), and improved precision and recall (IPR, Prec, and Rec) (Kynkäänniemi et al., 2019). For compression, we report PSNR and MS-SSIM (Wang et al., 2003) under different levels of bits per pixel (bpp).

## 5.1 MAIN RESULTS

**Image Reconstruction.** We first compare the image reconstruction result of BSQ on COCO and ImageNet ($256 \times 256$) with state-of-the-art image tokenizers, including DALL-E dVAE (Ramesh et al., 2021), SD-VAE 1.x (Rombach et al., 2022), SD-VAE 2.x, SDXL-VAE (Podell et al., 2023), MaskGIT (Chang et al., 2022), and ViT-VQGAN (Yu et al., 2022). To perform a comprehensive and fair comparison, we rerun all models using the same augmentation on COCO 2017val and ImageNet-1k val except the undisclosed ViT-VQGAN. From Table 1, we can see that our model outperforms prior works on all metrics (PSNR, SSIM, LPIPS, and rFID), often by a big margin. In Figure 4 and Figure 9, we show reconstructed images produced compared to the best prior work, SDXL-VAE. Our method preserves more details about high-frequency texture and fine-grained shape/geometry.

To compare the compression capability of different bottleneck modules, We study the **effective number of bits per token** (**# bits**). For VQ-based models, # bits equals to $\log_2(K)$, where $K$ is the codebook size; For KL-regularized models (SD-VAE 2.x and XL), since the latent is continuous, we count # bits as the latent dimension multiplied by the numeric precision (here we use 16 since the checkpoint is stored in FP16). For our BSQ, # bits is $L$ because each latent channel is binary. We summarize the key observations as follows. (1) BSQ efficiently compresses image patches into a small amount of bits. It reconstructs images better in all metrics using fewer bits per token than the second-best method (SDXL-VAE). (2) BSQ is also computationally efficient. Although the ViT-based backbone doubles the parameters, our method yields a $2.4\times$ higher throughput than SDXL-VAE. MaskGIT runs at a comparable speed but reconstructs significantly worse because of a small codebook size and more spatial downsampling. (3) BSQ is generalizable across different domains of images. ImageNet is relatively object-centric while COCO is more scene-centric. Though trained on ImageNet only, our method does well on the scene-centric COCO too. It even works better than SD-VAE 1./2.x trained on the similarly scene-centric OpenImages dataset (Kuznetsova et al., 2020).

**Video Reconstruction.** We present the video reconstruction on both UCF-101 training and validation subsets in Table 2. First, we use the image tokenizer to reconstruct the video frame by frame. BSQ works slightly better than VQ but neither is comparable to the specialized video tokenizers fine-tuned on video data shown in the lower half of Table 2. Second, we finetune the image tok-

Table 2: **Video reconstruction results on UCF-101 (split 1).**

| Method | Backbone | Quantizer | Param. | # bits | UCF-101 train PSNR↑ | SSIM↑ | LPIPS↓ | rFVD↓ | UCF-101 val PSNR↑ | SSIM↑ | LPIPS↓ | rFVD↓ |
|---|---|---|---|---|---|---|---|---|---|---|---|---|
| (IMAGE TOKENIZER, W/O ADAPTING TO VIDEOS) | | | | | | | | | | | | |
| Ours | ViT | VQ | 174M | 14 | 25.64 | .8142 | .1120 | 357 | 25.58 | .8120 | .1146 | 382 |
| Ours | ViT | BSQ | 174M | 18 | 25.86 | .8273 | .1089 | 326 | 25.83 | 0.8259 | 0.1108 | 342 |
| (IMAGE TOKENIZER → VIDEO TOKENIZER) | | | | | | | | | | | | |
| MaskGIT (Chang et al., 2022) | 2D CNN | VQ | 53M | 10 | 21.5 | .685 | 0.114 | 216 | - | - | - | - |
| TATS (Ge et al., 2022) | 3D CNN | VQ | 32M | 14 | - | - | - | 162 | | | | |
| MAGVIT-L (Yu et al., 2023) | 3D CNN | VQ | 158M | 10 | 22.0 | .701 | .0990 | 25 | - | - | - | - |
| MAGVIT-v2 (Yu et al., 2024) | C.-3D CNN | LFQ | 158M | 18 | - | - | .0694 | 16.12 | - | - | - | - |
| MAGVIT-v2 (Yu et al., 2024) | C.-3D CNN | LFQ | N/A (>158M) | 18 | - | - | .0537 | 8.62 | - | - | - | - |
| Ours | non-BC ViT | VQ | 174M | 14 | 33.06 | .9518 | .0223 | 9.16 | 32.92 | .9506 | .0228 | 12.79 |
| Ours | BC ViT | VQ | 174M | 14 | 32.81 | .9496 | .0236 | 10.76 | 32.68 | .9484 | .0241 | 14.17 |
| Ours | non-BC ViT | BSQ | 174M | 18 | 32.43 | .9479 | .0213 | 7.34 | 31.88 | .9410 | .0254 | 10.57 |
| Ours | BC ViT | BSQ | 174M | 18 | 32.08 | .9421 | .0244 | 8.08 | 31.49 | .9357 | .0276 | 11.62 |
| Ours | BC ViT | BSQ | 174M | 36 | **33.80** | **.9606** | **.0159** | **4.10** | **33.55** | **.9588** | **.0167** | **6.21** |

Table 4: **Visual compression results on Kodak and MCL-JCV** (640 × 360).

(a) Image Compression results on Kodak.

| Method | BPP | PSNR↑ | MS-SSIM (dB)↑ | LPIPS↓ |
|---|---|---|---|---|
| JPEG2000 | 0.2986 | 29.192 | 11.574 | .1892 |
| WebP | 0.2963 | **29.151** | 12.193 | .1655 |
| MAGVIT2 | 0.2812 | 23.467 | 8.103 | .1260 |
| VQ | 0.2812 | 26.987 | 12.580 | .0944 |
| Ours | **0.2812** | 27.785 | **12.852** | **.0823** |
| Ours (w/. AC) | 0.2073 | —"— | —"— | —"— |

(b) Video compression results on MCL-JCV.

| Method | BPP | PSNR↑ | MS-SSIM↑ | LPIPS↓ |
|---|---|---|---|---|
| MAGVIT | 0.0391 | 23.70 | 0.846 | .144 |
| MAGVIT-v2 | 0.0508 | 27.83 | 0.92 | .104 |
| H.264 | 0.1373 | 35.415 | 0.9796 | .0949 |
| H.265 | 0.1373 | **35.670** | 0.9807 | .0908 |
| Ours (w/o. AC) | 0.2333 | 33.698 | **0.9818** | **.0501** |
| Ours (w/. AC) | 0.1373 | 33.698 | **0.9818** | **.0501** |

enizer on videos and see significant improvements. For example, our 18-bit BSQ with causal ViT reduces rFVD from 342 to 11.62 and improves PSNR from 25.83 to 31.49 dB. The compared prior methods include: (1) MaskGIT (Chang et al., 2022) which is a fine-tuned 2D-CNN based tokenizer, (2) TATS (Ge et al., 2022) which uses a 3D CNN with replicated padding, (3) MAGVIT (Yu et al., 2022) whose 3D CNN is initialized by zero-inflating a 2D filter, and (4) MAGVIT-v2 (Yu et al., 2024) which makes 3D CNN causal. We take their reported numbers directly. Our models with all configurations outperform MAGVIT-v2 with a comparable number of parameters (174M *vs.* 158M) by a large margin. The best-performing MAGVIT-v2 uses a larger backbone and achieves a rFVD of 8.62. Our causal BSQ-ViT with $L = 18$ achieves an 8.08 rFVD and halves the LPIPS. For BSQ with $L = 36$, our method further improves the reconstruction metrics.

We also show the effect of using block-wise causal masks. The non-causal variant (non-BC) works slightly better on all metrics because now the model can look at all visual patches within the temporal context window. This result resembles the observations in video compression that using bidirectional predicted pictures (B-frames) benefits compression quality given the same group of pictures (GoP).

**Image Generation.** Our BSQ-ViT tokenizer can be seamlessly integrated into existing generative models for visual generation. We follow MaskGIT (Chang et al., 2022), a masked language modeling approach. Unlike MaskGIT with a VQ-VAE with $K = 1024$, BSQ-ViT has an effective vocabulary size of $2^L$ and $L = 18$, resulting in a slow embedding lookup. We fix it by dividing each token into groups and treating sub-tokens independently with

Table 3: **Image generation results on ImageNet-1K** (128 × 128). [†]The number is taken from the paper.

| Category | Method | # steps | FID↓ | IS↑ | Prec↑ | Rec↑ |
|---|---|---|---|---|---|---|
| GAN | BigGAN (Brock et al., 2018) | 1 | 6.02 | **145.8** | **0.86** | 0.35 |
| Diffusion | ADM (Dhariwal & Nichol, 2021) | 1,000 | 5.91 | 93.3 | 0.70 | **0.65** |
| Masked LM | VQ | 12 | [†]9.4 | - | - | - |
| | FSQ (Mentzer et al., 2024) | 12 | [†]8.5 | - | - | - |
| | BSQ (Ours) | 12 | 5.69 | 48.5 | 0.85 | 0.42 |
| | BSQ (Ours) | 32 | **5.44** | 139.6 | 0.80 | 0.50 |

a similar rationale in Section 4.1. We increase the number of decoding steps accordingly. Table 3 shows that the masked LM with BSQ outperforms those with VQ and FSQ reported in (Mentzer et al., 2024). Our method achieves comparable results with other generation paradigms such as GAN-based (Brock et al., 2018) and diffusion-based (Dhariwal & Nichol, 2021) approaches. We show qualitative results in Figure 5.

**Image and Video Compression.** We compare BSQ with two image compression standards including JPEG2000 and WebP on the Kodak image dataset in Table 4a. we can see that BSQ without entropy coding outperforms JPEG2000 and WebP in terms of MS-SSIM with a slightly smaller bpp. We show a detailed RD Curve in Figure 12 in Sec G.2. We compare the video compression result on MCL-JCV in Table 4b. More results with detailed discussions on MCL-JCV and UVG can be

Table 5: **Abalation studies on ImageNet-1k val 128×128.**

| Method | $\ell_2$-norm | # bits ($K \times d$ or $L$) | PSNR$_\uparrow$ | SSIM$_\uparrow$ | LPIPS$_\downarrow$ | rFID$_\downarrow$ | Code usage |
|---|---|---|---|---|---|---|---|
| | ✓ | 10 (1024×32) | 23.61$_{\pm 3.21}$ | .6873$_{\pm .1211}$ | .1214$_{\pm .0434}$ | 7.05 | 57.5% |
| VQ | ✓ | 14 (16384×8) | 25.76$_{\pm 3.46}$ | .7834$_{\pm .0988}$ | .0669$_{\pm .0282}$ | 4.27 | 100.0% |
| | ✓ | 16 (65536×8) | 25.67$_{\pm 3.36}$ | .7851$_{\pm .0962}$ | .0706$_{\pm .0283}$ | 6.61 | 100.0% |
| | ✓ | 10 | 24.11$_{\pm 3.25}$ | .7250$_{\pm .1121}$ | .0919$_{\pm .0338}$ | 4.51 | 100.0% |
| BSQ | ✓ | 14 | 25.26$_{\pm 3.31}$ | .7710$_{\pm .0992}$ | .0784$_{\pm .0293}$ | 4.60 | 99.8% |
| | ✓ | 18 | 25.97$_{\pm 3.37}$ | .7990$_{\pm .0906}$ | .0629$_{\pm .0261}$ | 2.66 | 93.8% |
| LFQ | ✗ | 18 | 18.58$_{\pm 2.10}$ | .4828$_{\pm .1340}$ | .2951$_{\pm .0806}$ | 30.7 | 0.6% |

Table 6: **Ablation studies of the loss design.**

(a) Leave-one-out ablations for training losses.

| $\mathcal{L}_{\text{commit}}$ | $\mathcal{L}_{\text{entropy}}$ | | $\mathcal{L}_{\text{LPIPS}}$ | rFID | Code |
|---|---|---|---|---|---|
| | $H(p(\boldsymbol{c}|\boldsymbol{u}))$ | $-H(\mathbb{E}[p(\boldsymbol{c}|\boldsymbol{u})])$ | | | usage |
| ✓ | ✓ | ✓ | ✓ | 2.95 | 45.6% |
| ✗ | ✓ | ✓ | ✓ | 2.83 | 93.8% |
| ✓ | ✗ | ✓ | ✓ | 2.44 | 78.3% |
| ✓ | ✓ | ✗ | ✓ | 13.8 | 13.3% |
| ✓ | ✓ | ✓ | ✗ | 19.2 | 6.9% |

(b) Group size. ($L = 18$)

| group size | rFID $\downarrow$ | Code usage | Speed (ms) |
|---|---|---|---|
| $g = 18$ | (OOM) | | 70.0 |
| $g = 9$ | 2.83 | 93.8% | 0.335 |
| $g = 6$ | 2.76 | 95.2% | 0.232 |
| $g = 3$ | 3.32 | 96.0% | 0.233 |
| Ours ($g = 1$) | 2.86 | 95.1% | 0.212 |

found in Figure 8 in Section G.3. Simply flattening the video token sequence to a bitstream achieves an MS-SSIM of 0.9818 at 0.2333 bpp. Although this is not great, we use an auto-regressive model to predict the conditional probability such that the bpp is reduced by 41%. This leads to a better tradeoff than standard video codecs including H.264 and HEVC.

## 5.2 ABLATION STUDIES

For ablation studies, we train an ImageNet image tokenizer with resolution 128×128 with $p = 8$, although our conclusions generally hold for higher resolution, *e.g.* 256×256 in Sec. 5.1.

**BSQ vs VQ.** Table 5 shows that BSQ and VQ follow a similar trend: better reconstruction for increased $L$. Since $K = 2^{18}$ results in an out-of-memory issue, we try a smaller $K = 2^{16} = 65536$ for VQ. The gain for VQ already diminishes even though the small bottleneck dimension of 8 still guarantees full code usage. In contrast, BSQ consistently works better on all metrics when $L = 18$.

**Importance of $\ell_2$ normalization in BSQ.** We remove the $\ell_2$ normalization in BSQ, which is equivalent to LFQ, and show results in the last rows of Table 5. We see much lower code usage and worse rFID, indicating that LFQ does not work well with a ViT-based tokenization encoder.

**Contribution of losses.** We study the effect of each loss in Table 6a. Although it is computationally prohibitive to enumerate all combinations of loss terms and their associative weights, we conduct a simple "leave-one-out" setting where one of the losses is removed at a time. BSQ works slightly better after removing $\mathcal{L}_{\text{commit}}$ and $H(p(\boldsymbol{c}|\boldsymbol{u}))$. However, the code usage varies greatly. The best configuration is to keep the minimal entropy term while dropping the commitment loss. The commit loss is unnecessary because of the strictly bounded quantization error in BSQ. On the contrary, the dataset entropy maximization term and perceptual term do matter. Without $-H(\mathbb{E}_{\boldsymbol{u}}[p(\boldsymbol{c}|\boldsymbol{u})])$, rFID increases to 13.8 while the code usage in the validation set significantly drops to 13.3%. We also observe that the perceptual loss is important for low FID and high code usage. However, a deeper look into its role is beyond the scope of this paper.

**Approximating the dataset entropy term.** We now show the efficacy of approximating the dataset entropy term using Equation 9. We compare with the approximation method in (Yu et al., 2024) that computes entropy in sub-groups of dimensions with varying group size $g \in \{9, 6, 3\}$. Our approximation method can also be interpreted as a group size of $g = 1$. From Table 6b, we conclude that our approximation achieves a similar level of rFID and code usage compared to other setups while running the fastest.

## 6 CONCLUSIONS

We present a new transformer-based image and video tokenizer with Binary Spherical Quantization (BSQ). The transformer-based architecture effortlessly integrates image and video tokenization over an arbitrary time horizon. The Binary Spherical Quantization allows for efficient and effective training of the quantized bottleneck. Our results indicate that the proposed tokenizer runs at a faster

speed, reconstructs with higher fidelity, and in combination with a sequence model offers a strong baseline for lossy video compression and image synthesis.

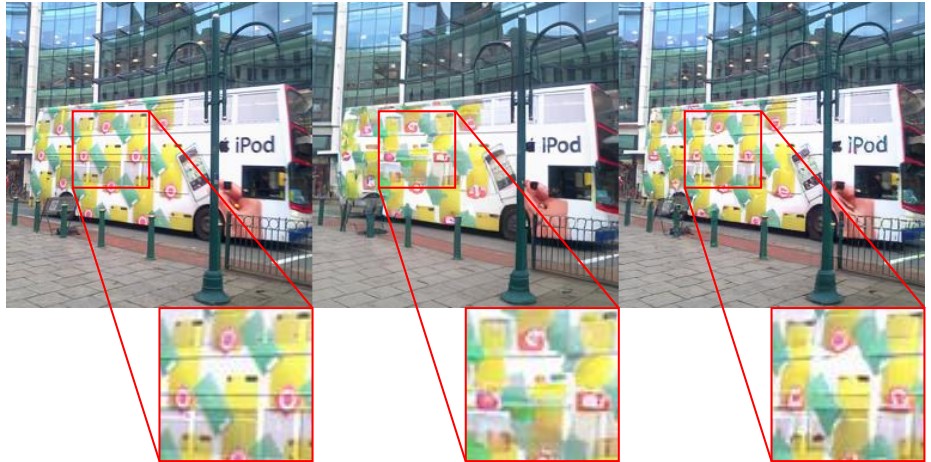

Figure 4: Reconstruction results of BSQ-ViT (**right**) compared to the original image (**left**) and SDXL-VAE (Podell et al., 2023) (**middle**). See Figure 9 for more visualization.

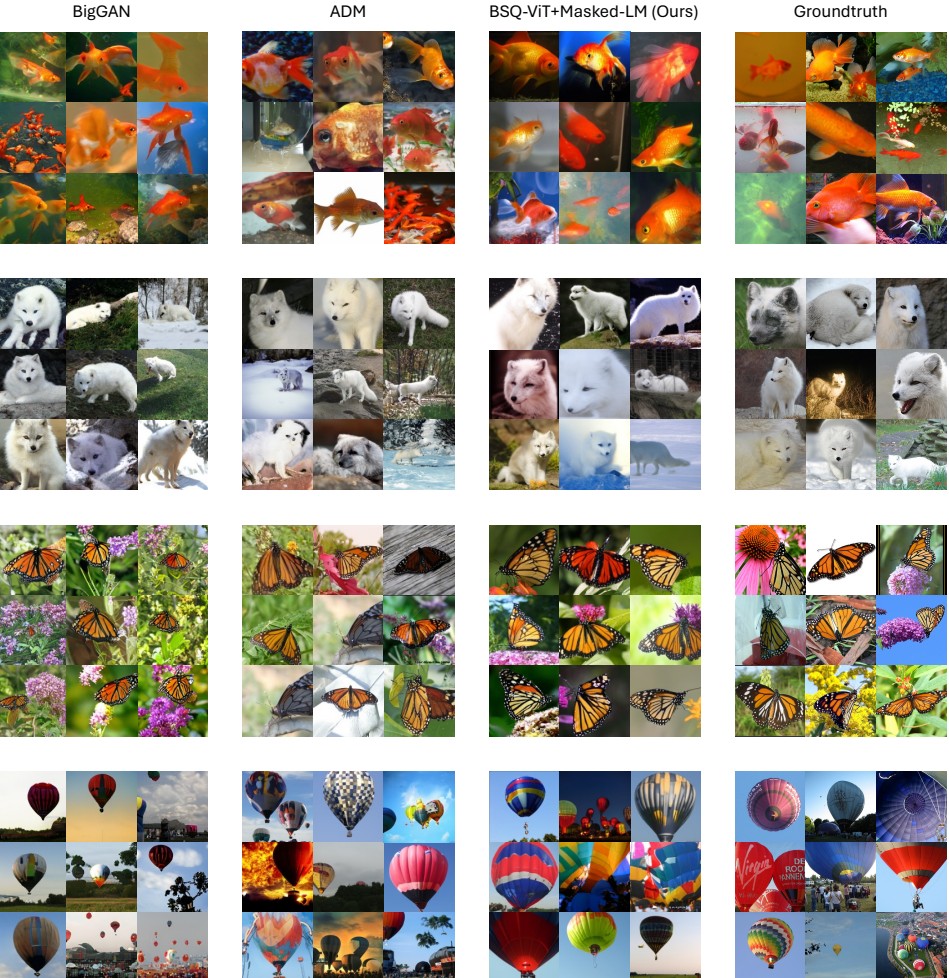

Figure 5: Sampled generation results of BSQ-ViT + Masked-LM (**second column from left**) compared to BigGAN (Brock et al., 2018) (**right**), ADM (Dhariwal & Nichol, 2021) (**second column from right**) and the original images (**left**). Classes are 1: goldfish, 279: arctic fox, 323: monarch butterfly, 417: balloon.

ACKNOWLEDGMENTS

This material is based upon work in part supported by the National Science Foundation under Grant No. IIS-1845485.

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

Table 7: **Comparing BSQ and LFQ.**

|  | LFQ (Yu et al., 2024) | BSQ (Ours) |
|---|---|---|
| Quantized output | $\hat{\boldsymbol{v}} = \text{sign}(\boldsymbol{v})$ | $\hat{\boldsymbol{u}} = \frac{1}{\sqrt{L}}\text{sign}(\boldsymbol{u}) = \frac{1}{\sqrt{L}}\text{sign}(\frac{\boldsymbol{v}}{|\boldsymbol{v}|})$ |
| STE gradient | $\frac{\partial \hat{v}_i}{\partial v_i} = 1$ | $\frac{\partial \hat{u}_i}{\partial v_i} = \frac{1}{\sqrt{L}}(1 - v_i^2/|\boldsymbol{v}|^2)$ |
| Quantization Error | $\mathbb{E}_{\boldsymbol{v}}[d(\boldsymbol{v}, \hat{\boldsymbol{v}})] = \infty$ | $\mathbb{E}_{\boldsymbol{u}}[d(\boldsymbol{u}, \hat{\boldsymbol{u}})] < \sqrt{2 - \frac{2}{\sqrt{L}}} < \sqrt{2}$ |
|  | Unbounded | Upper-bounded (See Section C.4) |
| Training objective | $\mathcal{L}_{\text{MSE}}, \mathcal{L}_{\text{commit}}, \mathcal{L}_{\text{LPIPS}}, \mathcal{L}_{\text{GAN}},$ | $\mathcal{L}_{\text{MSE}}, \mathcal{L}_{\text{LPIPS}}, \mathcal{L}_{\text{GAN}},$ |
|  | $\mathcal{L}_{\text{entropy}} = H[p(\boldsymbol{c}|\boldsymbol{v})] - H[\mathbb{E}_{\boldsymbol{v}}[p(\boldsymbol{c}|\boldsymbol{v})]]$ | $\mathcal{L}_{\text{entropy}} = H[p(\boldsymbol{c}|\boldsymbol{u})] - \hat{H}[\mathbb{E}_{\boldsymbol{u}}[p(\boldsymbol{c}|\boldsymbol{u})]]$ |

Lijun Yu, José Lezama, Nitesh B Gundavarapu, Luca Versari, Kihyuk Sohn, David Minnen, Yong Cheng, Agrim Gupta, Xiuye Gu, Alexander G Hauptmann, et al. Language model beats diffusion–tokenizer is key to visual generation. In *ICLR*, 2024.

Richard Zhang, Phillip Isola, Alexei A Efros, Eli Shechtman, and Oliver Wang. The unreasonable effectiveness of deep features as a perceptual metric. In *CVPR*, 2018.

Chuanxia Zheng and Andrea Vedaldi. Online clustered codebook. In *ICCV*, 2023.

Jinghao Zhou, Chen Wei, Huiyu Wang, Wei Shen, Cihang Xie, Alan Yuille, and Tao Kong. iBOT: Image BERT pre-training with online tokenizer. In *ICLR*, 2022.

# A ARITHMATIC CODING DETAILS

Starting from the initial interval $I_0 = [0, 1)$, the AC encoder recursively partitions the interval into a series of sub-interval $I_n = [l_n, u_n)$ such that $I_n \subset I_{n-1} \subset \cdots \subset I_0$, and $I_n$ is determined by $I_{n-1}$ and $\rho(y|x_{<n})$.

$$I_n(y) = \left[l_{n-1} + (u_{n-1} - l_{n-1}) \sum_{y=1}^{x_n - 1} \rho(y|x_{<n}), \quad l_{n-1} + (u_{n-1} - l_{n-1}) \sum_{y=1}^{x_n} \rho(y|x_{<n})\right). \quad (11)$$

Any number in the final interval $I_N$ can sufficiently represent the encoded sequence. To obtain the final bit stream, we take a binary fraction $\lambda = \sum_{i=1}^{C} b_i \times 2^{-i}$, $b_i \in \{0, 1\}$ in $I_N$ such that $l_N \le \lambda < u_N$. The bit stream $\{b_0, \ldots, b_C\}$ is the encoding result with a length of $C$ bits.

The AC decoder takes in $\lambda$, starts with $I_0$, and performs a similar interval partitioning process. At the $n$-th step, the decoder queries the model $\rho_n(y|x_{<n})$, calculate the sub-intervals for all possible values of $y$ using Equation 11, and decodes output $x_n$ that leads to $\lambda \in I_n(x_n)$. The decoder can recover the encoded token sequence by continuing with $I_{n+1}$ based on the decoded $x_n$ and repeating for step $n + 1$ for $N$ steps.

In practice, the encoder and the decoder can be implemented efficiently with fixed-length integer numbers and operate incrementally for arbitrarily long input sequences.

# B COMPARISON BETWEEN BSQ AND LFQ

In Section 4.1, we have introduced the mechanism of BSQ and briefly discussed the connections and differences with LFQ. We summarize them in Table 7. Note that STE gradient in BSQ is anisotropic and is more likely to be a good estimation because of an upper-bounded quantization error regardless of $L$. This property explains why a commitment loss like $\mathcal{L}_{\text{commit}}(\hat{\boldsymbol{u}}, \boldsymbol{u})$ is not needed in BSQ but useful for LFQ.

# C PROOFS

## C.1 PROOF OF EQUATION 7

Before proving Equation 7, we will first prove the following identity:

Let $\boldsymbol{u} \in \mathbb{R}^L$, $\boldsymbol{C} = \Omega^L \in \mathbb{R}^{L \times 2^L}$ for $\Omega = \{-\frac{1}{\sqrt{L}}, \frac{1}{\sqrt{L}}\}$,

$$\sum_{\boldsymbol{c} \in \boldsymbol{C}} e^{\tau \boldsymbol{u}^\top \boldsymbol{c}} = \sum_{\boldsymbol{c} \in \boldsymbol{C}} \prod_{d=1}^{L} e^{\tau u_d c_d} = \prod_{d=1}^{L} \sum_{c_d \in \Omega} e^{\tau u_d c_d}. \tag{12}$$

**Proof.** With $\tau$ dropped for simplicity of notation.

$$\sum_{\boldsymbol{c} \in \boldsymbol{C}} e^{\boldsymbol{u}^\top \boldsymbol{c}} = \sum_{\boldsymbol{c} \in \boldsymbol{C}} \prod_{k=1}^{L} e^{u_k c_k}$$

$$= \sum_{c_1 \in \Omega} \sum_{c_2 \in \Omega} \cdots \sum_{c_L \in \Omega} \prod_{d=1}^{L} e^{u_d c_d}$$

$$= \sum_{c_1 \in \Omega} \sum_{c_2 \in \Omega} \cdots \sum_{c_L \in \Omega} e^{u_L c_L} \prod_{d=1}^{L-1} e^{u_d c_d}$$

$$= \sum_{c_1 \in \Omega} \sum_{c_2 \in \Omega} \cdots \sum_{c_{L-1} \in \Omega} \left( \prod_{d=1}^{L-1} e^{u_d c_d} \right) \left( \sum_{c_L \in \Omega} e^{u_L c_L} \right)$$

$$= \ldots$$

$$= \left( \sum_{c_1 \in \Omega} e^{u_L c_L} \right) \left( \sum_{c_2 \in \Omega} e^{u_2 c_2} \right) \cdots \left( \sum_{c_L \in \Omega} e^{u_L c_L} \right) = \prod_{d=1}^{L} \sum_{c_d \in \Omega} e^{u_d c_d}. \square$$

Therefore, the probability of $\boldsymbol{u}$ being assigned to $\boldsymbol{c}_i$ can be written as:

$$\hat{q}(\hat{\boldsymbol{c}}|\boldsymbol{u}) = \frac{e^{\tau \boldsymbol{u}^\top \hat{\boldsymbol{c}}}}{\sum_{\boldsymbol{c} \in \boldsymbol{C}} e^{\tau \boldsymbol{u}^\top \boldsymbol{c}}} = \frac{\prod_{d=1}^{L} e^{\tau u_d \hat{c}_d}}{\prod_{d=1}^{L} \sum_{c_d \in \{-\frac{1}{\sqrt{L}}, \frac{1}{\sqrt{L}}\}} e^{\tau u_d c_d}} \qquad \text{(Using Equation 12)}$$

$$= \prod_{d=1}^{L} \frac{e^{\tau u_d \hat{c}_d}}{e^{\tau u_d \hat{c}_d} + e^{-\tau u_d \hat{c}_d}} \qquad \text{(since } c_d = \pm\frac{1}{\sqrt{L}} = \pm\hat{c}_d)$$

$$= \prod_{d=1}^{L} \sigma(2\tau u_d \hat{c}_d).$$

## C.2 PROOF OF EQUATION 8

Since $\hat{q}(\hat{\boldsymbol{c}}|\boldsymbol{u}) = \prod_{d=1}^{L} \sigma(2\tau u_d \hat{c}_d)$ each variable $c_d$ is independent of each other. Thus by definition

$$H[\hat{q}(\boldsymbol{c}|\boldsymbol{u})] = \sum_{d=1}^{L} H(\sigma(2\tau u_d c_d)). \quad \square$$

## C.3 PROOF OF EQUATION 9

Now we look at $H[\mathbb{E}_{\boldsymbol{u}}[\hat{q}(\boldsymbol{c}|\boldsymbol{u})]]$. We first compute $Q(\boldsymbol{c}) = \mathbb{E}_{\boldsymbol{u}}[\hat{q}(\boldsymbol{c}|\boldsymbol{u})]$.

$$Q(\boldsymbol{c}) = \mathbb{E}_{\boldsymbol{u}}[\hat{q}(\boldsymbol{c}|\boldsymbol{u})] = \frac{1}{N} \sum_{\boldsymbol{u}} \hat{q}(\boldsymbol{c}|\boldsymbol{u}) = \frac{1}{N} \sum_{\boldsymbol{u}} \prod_{k}^{L} \sigma(2 u_k c_k).$$

Unlike $\boldsymbol{c}$, $\boldsymbol{u}$ does not factorize like Equation 12. This would require us to compute $Q(\boldsymbol{c})$ as a full distribution over $2^L$ states, which is slow ($O(L \times 2^L)$) and easily overfits. Instead, we approximate $Q(\boldsymbol{c})$ by a factorized distribution $\tilde{q}(\boldsymbol{c}) = \prod_{d=1}^{L} \tilde{q}_d(c_d)$, where $c_d \in \Omega$ for $\Omega = \{-\frac{1}{\sqrt{L}}, \frac{1}{\sqrt{L}}\}$, using

an M-projection. We again omit $\tau$ for notational brevity.

$$D(Q\|\tilde{q}) = H(Q, \tilde{q}) - H(Q)$$

$$= -\sum_{i=1}^{2^L} Q(\boldsymbol{c}_i) \log \tilde{q}(\boldsymbol{c}_i) - H(Q)$$

$$= -\sum_{i=1}^{2^L} Q(\boldsymbol{c}_i) \sum_d \log \tilde{q}_d(c_d) - H(Q)$$

$$= -\sum_d \underbrace{\sum_{i=1}^{2^L} Q(\boldsymbol{c}_i) \log \tilde{q}_d(c_d)}_{} - H(Q)$$

$$= -\sum_d \left( \log \tilde{q}_d(\boldsymbol{c}_d = 1) \sum_{\boldsymbol{c}_{-d}} Q(\boldsymbol{c}_i) + \log \tilde{q}_d(c_d = -1) \sum_{\boldsymbol{c}_{-d}} Q(c_i) \right) - H(Q)$$

$$= -\sum_d \sum_{c_d \in \{-1,1\}} \log \tilde{q}_d(c_d) \sum_{\boldsymbol{c}_{-d}} Q(\boldsymbol{c}) - H(Q), \text{ where } \boldsymbol{c}_{-d} \text{ sums over all dimensions except } d.$$

The minimizer of the above projection $\frac{\partial}{\partial \tilde{q}_d} D(Q\|\tilde{q}) = 0$

$$\tilde{q}_d(c_d)^* = \sum_{\boldsymbol{c}_{-d}} Q(\boldsymbol{c}) = \mathbb{E}_{\boldsymbol{u}} \left[ \sum_{\boldsymbol{c}_{-d}} \hat{q}(\boldsymbol{c}|\boldsymbol{u}) \right]$$

$$= \mathbb{E}_{\boldsymbol{u}} \left[ \sum_{\boldsymbol{c}_{-d}} \prod_k \sigma(2u_k c_k) \right] = \mathbb{E}_{\boldsymbol{u}} \left[ \sum_{\boldsymbol{c}_{-d}} \sigma(2u_d c_d) \prod_{k \neq d} \sigma(2u_k c_k) \right]$$

$$= \mathbb{E}_{\boldsymbol{u}} \left[ \sigma(2u_d c_d) \sum_{\boldsymbol{c}_{-d}} \prod_{k \neq d} \sigma(2u_k c_k) \right] = \mathbb{E}_{\boldsymbol{u}} \left[ \sigma(2u_d c_d) \underbrace{\prod_{k \neq d} \sum_{\boldsymbol{c}_{-d}} \sigma(2u_k c_k)}_{=1} \right] = \mathbb{E}_{\boldsymbol{u}} \left[ \sigma(2u_d c_d) \right]$$

Therefore, the entropy term is simplified to:

$$H(\tilde{q}) = \sum_d H(\tilde{q}_d(c_d)) = \sum_d H\left( \mathbb{E}_{\boldsymbol{u}}[\sigma(2u_d c_d)] \right).$$

By the nature of the above derivation the cross entropy $H(Q, \tilde{q}) = H(\tilde{q})$ equals the entropy of the approximation. This means that $D(Q\|\tilde{q}) = H(\tilde{q}) - H(Q) \geq 0$, and the entropy of the approximation is an upper bound $H(\tilde{q}) \geq H(Q)$ to the true entropy.

In practice, this bound is relatively tight. The most adversarial distribution $P(\boldsymbol{u})$ is $P(\frac{1}{\sqrt{L}}\vec{1}) = \frac{1}{2}$ and $P(-\frac{1}{\sqrt{L}}\vec{1}) = \frac{1}{2}$, where all inputs are maximally correlated, but the factorized distribution is not. Figure 6 shows an empirical estimate of this approximation error for various values of $\tau$. In practice, we use $\tau = \frac{1}{100}$, which has little to no approximation error.

## C.4 PROOF OF THE QUANTIZATION ERROR BOUND OF BSQ (EQUATION 10)

We consider $\ell_2$-distance $d(\boldsymbol{u}, \hat{\boldsymbol{u}}) = \|\boldsymbol{u} - \hat{\boldsymbol{u}}\|$. A simple (but loose) bound is:

$$\mathbb{E}_{\boldsymbol{u}}[d(\boldsymbol{u}, \hat{\boldsymbol{u}})] = \mathbb{E}_{\boldsymbol{u}}[d_{\max}(\boldsymbol{u}, \hat{\boldsymbol{u}})] < \sqrt{2 - \frac{2}{\sqrt{L}}} < \sqrt{2}, \tag{13}$$

where $d_{\max}$ is attained if $\boldsymbol{u}$ is at any axis, $\boldsymbol{u} = [\underbrace{0, \cdots, 0}_{n}, 1, \underbrace{0, \cdots, 0}_{L-1-n}]$.

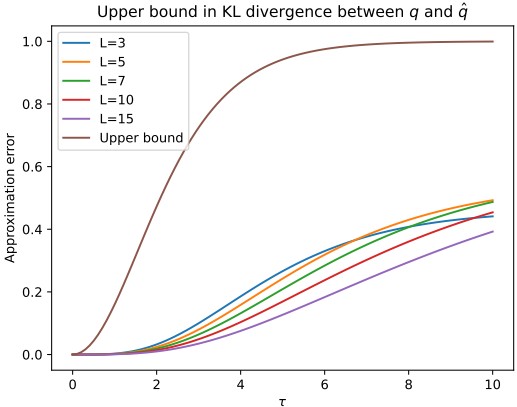

Figure 6: Empirical estimation of the approximation error with respect to $\tau$ at different bottleneck dimensions $L$.

**Proof of Equation 13.** The sketch of derivation to $d_{max}(\boldsymbol{u}, \hat{\boldsymbol{u}})$ is as follows. For an arbitrary vector $\boldsymbol{u}$ residing on the unit hypersphere ($|\boldsymbol{u}|_2 = 1$) and its quantized counterpart $\hat{\boldsymbol{u}}$ also on the unit hypersphere, we can write the quantization error as $d(\boldsymbol{u}, \hat{\boldsymbol{u}}) = \sqrt{2 - 2\boldsymbol{u} \cdot \hat{\boldsymbol{u}}}$. The maximum of $d(\boldsymbol{u}, \hat{\boldsymbol{u}})$ is therefore attained when the minimum of $\boldsymbol{u} \cdot \hat{\boldsymbol{u}} = \sum u_i \hat{u}_i$ is achieved, where $u_i$ and $\hat{u}_i$ are elements of $\boldsymbol{u}$ and $\hat{\boldsymbol{u}}$, respectively. In BSQ the quantization law is that $\hat{u}_i = \frac{1}{\sqrt{L}}\text{sign}(u_i)$. This leads to $u_i \hat{u}_i = \frac{1}{\sqrt{L}}|u_i|$ and $\boldsymbol{u} \cdot \hat{\boldsymbol{u}} = \frac{\sum |u_i|}{\sqrt{L}} = \frac{|\boldsymbol{u}|_1}{\sqrt{L}}$. Minimizing $\boldsymbol{u} \cdot \hat{\boldsymbol{u}}$ is then equivalent to finding an $\ell_2$ unit vector with the minimal $\ell_1$ norm, which is known to be achieved only if the vector $\boldsymbol{u}$ is a one-hot vector and the minimal $\ell_1$ norm is 1. This leads to the maximum of $d(\boldsymbol{u}, \hat{\boldsymbol{u}})$ being $\sqrt{2 - \frac{2}{\sqrt{L}}}$.

To achieve a tighter bound, we first expand the definition,

$$\mathbb{E}_{\boldsymbol{u}}\left[d(\boldsymbol{u}, \hat{\boldsymbol{u}})\right] = \frac{\overbrace{\int \cdots \int}^{S^{L-1}} d_{S^{L-1}}V d(\boldsymbol{u}, \hat{\boldsymbol{u}})}{\underbrace{\int \cdots \int}_{S^{L-1}} d_{S^{L-1}}V}, \tag{14}$$

where $S^{L-1} = \{x \in \mathbb{R}^L : \|x\| = 1\}$ denotes the unit $L$-sphere of radius 1 and $d_{S^{L-1}}V$ denotes its surface area element. We further define a hyperspherical coordinate system that is analogous to the spherical coordinate system for 3D Euclidean space or the polar coordinate system for 2D space to represent the surface area element.

$$u_1 = \cos(\varphi_1),$$
$$u_2 = \sin(\varphi_1)\cos(\varphi_2),$$
$$u_3 = \sin(\varphi_1)\sin(\varphi_2)\cos(\varphi_3),$$
$$\cdots$$
$$u_{L-1} = \sin(\varphi_1)\sin\varphi_2 \cdots \sin(\varphi_{L-2})\cos(\varphi_{L-1}),$$
$$u_L = \sin(\varphi_1)\sin\varphi_2 \cdots \sin(\varphi_{L-2})\sin(\varphi_{L-1}),$$

$$\text{(surface area element) } d_{S^{L-1}}V = \sin^{L-2}(\varphi_1)\sin^{L-3}(\varphi_2)\cdots\sin(\varphi_{L-2})d\varphi_1 \cdots d\varphi_{L-1},$$

$$\text{(surface area) } S_{L-1} = \underbrace{\int \cdots \int}_{S^{L-1}} d_{S^{n-1}}V = \frac{2\pi^{L/2}}{\Gamma(\frac{L}{2})}.$$

Due to symmetry, we assume the subarea $A^{L-1}$ where $\forall i \in \{1, \cdots, L\}, u_i > 0$, and it will be quantized to $\boldsymbol{c}_1 = \hat{\boldsymbol{u}}_1 = \frac{1}{\sqrt{L}}\overrightarrow{1}$. The unit hypersphere $S^{L-1}$ has $2^L$ of such subareas interchangeably.

Computing Equation 14 is equivalent to

$$\mathbb{E}_{\boldsymbol{u}}\left[d(\boldsymbol{u}, \hat{\boldsymbol{u}})\right] = \frac{\underbrace{\int \cdots \int}_{A^{L-1}} d_{S^{L-1}} V d(\boldsymbol{u}, \hat{\boldsymbol{u}})}{\underbrace{\int \cdots \int}_{A^{L-1}} d_{S^{L-1}} V}. \tag{15}$$

We expand the the numerator in Equation 15 as follows:

$$
\begin{aligned}
= \int_0^{\frac{\pi}{2}} \cdots \int_0^{\frac{\pi}{2}} & d_{S^{L-1}} V \{ [\cos(\varphi_1) - \frac{1}{\sqrt{L}}]^2 + [\sin(\varphi_1)\cos(\varphi_2) - \frac{1}{\sqrt{L}}]^2 + \cdots \\
& + [\sin(\varphi_1)\sin(\varphi_2)\cdots\sin(\varphi_{L-2})\cos(\varphi_{L-1}) - \frac{1}{\sqrt{L}}]^2 \\
& + [\sin(\varphi_1)\sin(\varphi_2)\cdots\sin(\varphi_{L-2})\sin(\varphi_{L-1}) - \frac{1}{\sqrt{L}}]^2 \}^{\frac{1}{2}}
\end{aligned}
\tag{16}
$$

It is composed of $L$ square terms. It is easy to see that the sum of constant terms leads to 1. Next, let's sum over all quadratic terms and keep on using $\sin^2(\theta) + \cos^2(\theta) = 1$:

$$\cos^2(\varphi_1) + \sin^2(\varphi_1)\cos^2(\varphi_2) + \cdots + \prod_{j=1}^{L-2}\sin^2(\varphi_j)\sin^2(\varphi_{L-1}) + \prod_{j=1}^{L-2}\sin^2(\varphi_j)\cos^2(\varphi_{L-1}) = 1$$

So the distance function to be integrated simplifies to

$$
\begin{aligned}
& \left[ 2 - \frac{2}{\sqrt{L}}\cos(\varphi_1) - \frac{2}{\sqrt{L}}\sin(\varphi_1)\cos(\varphi_2) - \cdots - \frac{2}{\sqrt{L}}\prod_{j=1}^{L-2}\sin(\varphi_j)\cos(\varphi_{L-1}) - \frac{2}{\sqrt{L}}\prod_{j=1}^{L-2}\sin(\varphi_j)\sin(\varphi_{L-1}) \right]^{\frac{1}{2}} \\
& < \left( 2 - \frac{2}{\sqrt{L}}\cos(\varphi_1) \right)^{\frac{1}{2}}.
\end{aligned}
$$

Plug into the numerator in Equation 15 and continue simplifying:

$$\underbrace{\int \cdots \int}_{A^{L-1}} d_{S^{L-1}} V \left( 2 - \frac{2}{\sqrt{L}}\cos(\varphi_1) \right)^{\frac{1}{2}} \tag{17}$$

$$= \underbrace{\underbrace{\int \cdots \int}_{A^{L-1}} d_{S^{L-2}} V}_{\frac{S_{L-2}}{2^{L-1}}} \int_0^{\frac{\pi}{2}} \left( 2 - \frac{2}{\sqrt{L}}\cos(\varphi_1) \right)^{\frac{1}{2}} \sin^{L-2}(\varphi_1) d\varphi_1. \tag{18}$$

Therefore, we have

$$\mathbb{E}_{\boldsymbol{u}}\left[d(\boldsymbol{u}, \hat{\boldsymbol{u}})\right] < \frac{2\Gamma(\frac{L}{2})}{\sqrt{\pi}\Gamma(\frac{L-1}{2})} \int_0^{\frac{\pi}{2}} \left( 2 - \frac{2}{\sqrt{L}}\cos(\varphi_1) \right)^{\frac{1}{2}} \sin^{L-2}(\varphi_1) d\varphi_1, \tag{19}$$

where RHS can be numerically computed and plotted in Figure 7.

## D   DATASET OVERVIEW

**ImageNet-1k** has 1.28M training images and 50,000 validation images; **COCO 2017val** has 5,000 images.

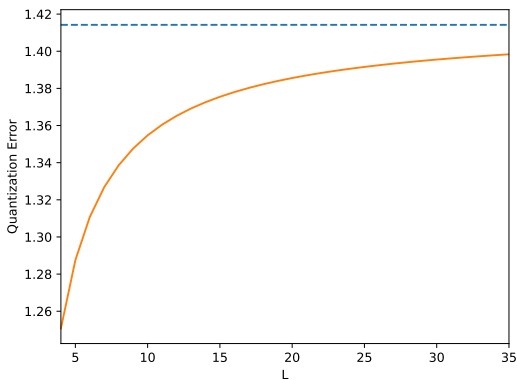

Figure 7: Quantization error with vocabulary size $L$.

**UCF101** has 13,320 video clips and three train-val splits. Following prior works (Yu et al., 2023), we consider split-1 which has 9,537 clips for training and 3,783 for validation.

The **MCL-JCV** dataset (Wang et al., 2016) consists of thirty 1080P (1,920×1,080) video sequences with 24∼30 FPS. The Open Ultra Video Group (**UVG**) dataset (Mercat et al., 2020) consists of sixteen 4K (3,840×2,160) test video sequences captured at 50/120 FPS. Following prior works (Agustsson et al., 2020), we report the performance on a subset of seven videos in YUV 8bit format at 120 FPS under the resolution of 1,920×1,080.

## E  IMPLEMENTATION DETAILS

**Training Image Tokenizers.** We train the image tokenizer with a batch size of 32 per GPU. We use AdamW optimizer (Loshchilov & Hutter, 2019) with $(\beta_1, \beta_2) = (0.9, 0.99)$ with $1 \times 10^{-4}$ weight decay. The base learning rate is $4 \times 10^{-7}$ (or a total learning rate of $1 \times 10^{-4}$) and follows a half-period cosine annealing schedule. The model is trained for 1M steps which amounts to 200 epochs over the entire ImageNet-1k training set. We did not heavily study the effect of loss weights. Instead, we keep $\gamma = 1$ in the entropy terms. We use a perceptual loss weight of 0.1 and an adversarial loss weight of 0.1 throughout the experiments.

**Evaluating Image Tokenizers.** We observe that reconstruction metrics vary with many factors, especially preprocessing (*e.g.* interpolation), input resolution, and downsample scales (Section D.2 in (Rombach et al., 2022)). we resize all images such that the smaller edge is 256 pixels using Lánczos interpolation in Table 1 and bilinear interpolation in Table 8, take the center crop ($H \times W$) = ($256 \times 256$), and ensure all models have the same spatial downsample ratio of $p = 8$ (except for MaskGIT, $p = 16$). We rerun all models on COCO 2017val and ImageNet-1k val except the undisclosed ViT-VQGAN.

**Training Video Tokenizers.** We finetune the video tokenizer with a batch size of 32 per GPU. The optimization schedule follows the image-based one but trains for fewer iterations. The network is initialized from the ImageNet-pretraining checkpoint and undergoes another 500K steps which amounts to 1600 epochs over UCF-101 split-1 train.

**Training a Masked Language Model for Generation.** The masked LM is a standard post-LN Transformer with 24 layers and a hidden dimension of 768 following MaskGIT (Chang et al., 2022). We train the masked LM on 2 nodes of $8\times$ GPUs (16 in total) with a total batch size of 1024 for 1M steps. We use AdamW optimizer with $(\beta_1, \beta_2) = (0.9, 0.96)$ with 0.045 weight decay. At inference time, we use a cosine unmasking schedule in MaskGIT (Chang et al., 2022) and set the sampling temperature to 15. We use classifier-free guidance (Ho & Salimans, 2022): At training, we replace 20% of the class condition labels with the mask token so that the model learns an unconditional distribution simultaneously. Let $\ell_c$ be class-conditioned logits and $\ell_\emptyset$ be unconditional logits. During inference, we interpolate logits using $\ell' = \ell_c + \alpha(\ell_c - \ell_\emptyset)$, where $\alpha = 0.5$.

**Training an Auto-Regressive Model for Arithmetic Coding.** The auto-regressive model is a Transformer with 24 layers and a hidden dimension 768. We train this model on $8\times$ GPUs with a total batch size of 64. We use AdamW optimizer with $(\beta_1, \beta_2) = (0.9, 0.96)$ with 0.045 weight decay.

Table 8: **Image reconstruction results on COCO2017 and ImageNet-1K** ($256 \times 256$). The settings strictly follow Table 1 except that all images are resized with **bilinear** interpolation.

| Method | Data | Arch. | Quant. | Param. | # bits | TP↑ | COCO2017 val | | | | ImageNet-1k val | | | |
|---|---|---|---|---|---|---|---|---|---|---|---|---|---|---|
| | | | | | | | PSNR↑ | SSIM↑ | LPIPS↓ | rFID↓ | PSNR↑ | SSIM↑ | LPIPS↓ | rFID↓ |
| DALL-E dVAE (Ramesh et al., 2021) | CC+YF | C | VQ | 98M | 13 | 34.0 | 26.97 | .0837 | .2544 | 48.60 | 27.31 | .7943 | .2544 | 32.63 |
| | | | | | | | ±3.41 | ±.0922 | ±.1057 | | ±3.81 | ±.1114 | ±.1057 | |
| MaskGIT (Chang et al., 2022) | IN-1k | C | VQ | 54M | 10 | 37.5 | 18.21 | .4596 | .1930 | 8.47 | 18.63 | .4619 | .1884 | 1.98 |
| | | | | | | | ±2.74 | ±0.1606 | ±.0444 | | ±2.90 | ±.1812 | ±.0497 | |
| ViT-VQGAN (Yu et al., 2022) | IN-1k | T-B | VQ | 182M | 13 | [†]7.5 | - | - | - | - | - | - | - | [†]1.55 |
| SD-VAE 1.x (Rombach et al., 2022) | OImg | C | VQ | 68M | 10 | 22.4 | 23.29 | .6705 | .0949 | 6.49 | 23.65 | .6615 | .0940 | 1.40 |
| | | | | | | | ±3.34 | ±.1316 | ±.0313 | | ±3.69 | ±.1540 | ±.0367 | |
| SD-VAE 1.x (Rombach et al., 2022) | OImg | C | VQ | 68M | 14 | 22.4 | 24.17 | .7042 | .0814 | 5.75 | 24.48 | .6931 | .0814 | 1.13 |
| | | | | | | | ±3.50 | ±.1276 | ±.0289 | | ±3.98 | ±.1502 | ±.0289 | |
| SD-VAE 1.x (Rombach et al., 2022) | OImg | C | KL | 68M | 64 | 22.4 | 23.21 | .6930 | .0908 | 5.94 | 23.54 | .6835 | .0899 | 1.22 |
| | | | | | | | ±3.24 | ±.1249 | ±.04282 | | ±3.62 | ±.1465 | ±.0337 | |
| SD-VAE 2.x (Podell et al., 2023) | OImg+ LAION | C | KL | 84M | 64 | 18.9 | 26.62 | .7722 | .0584 | 4.26 | 26.90 | .7592 | .0609 | 0.70 |
| | | | | | | | ±3.64 | ±.1086 | ±.0273 | | ±4.09 | ±.1300 | ±.0349 | |
| SDXL-VAE (Podell et al., 2023) | OImg+ LAION+? | C | KL | 84M | 64 | 18.9 | 27.08 | .7953 | .0541 | 3.93 | 27.37 | .7814 | .0574 | 0.67 |
| | | | | | | | ±3.88 | ±.1066 | ±.0250 | | ±4.36 | ±.1282 | ±.0320 | |
| Ours | IN-1k | T-B | BSQ | 174M | 18 | **45.1** | 26.89 | .8133 | .0652 | 5.41 | 27.78 | .8171 | .0633 | 0.99 |
| | | | | | | | ±3.47 | ±.0851 | ±.0255 | | ±3.99 | ±.0987 | ±.0307 | |
| Ours | IN-1k | T-B | BSQ | 174M | 36 | **45.1** | 29.85 | .8862 | .0341 | 3.07 | 30.12 | .8803 | .0355 | **0.36** |
| | | | | | | | ±3.65 | ±.0570 | ±.0163 | | ±4.13 | ±.0670 | ±.0207 | |
| Ours (w/. EMA) | IN-1k | T-B | BSQ | 174M | 36 | **45.1** | **30.19** | **.8904** | **.0314** | 3.07 | **30.45** | **.8843** | **.0329** | 0.42 |
| | | | | | | | ±3.69 | ±.0561 | ±.0153 | | ±4.19 | ±.0661 | ±.0194 | |

**Hardware.** The hardware for training is $8\times$GPU-servers with NVIDIA A5000 (24GB). Pre-training an image tokenizer and fine-tuning a video tokenizer in the full schedule is done across two servers with distributed training and takes around 5 days. Training the AR model for AC is done on an $8\times$GPU server and takes around 1 week. When measuring the tokenizer's throughput and the compression runtime, we use a server with $4\times$ A5000 GPU and $1\times$ AMD Ryzen Threadripper PRO 5975WX 32-Core CPU (64 threads).

## F  BASELINES FOR VIDEO COMPRESSION

Following SSF (Agustsson et al., 2020), we used FFmpeg[1] to produce the evaluation metrics for H.264 and HEVC. We use the commands provided in CompressAI (Bégaint et al., 2020).

```
ffmpeg -y -s:v $RESOLUTION -i $FILE.yuv -c:v h264 -crf $CRF \
-preset medium -bf 0 -pix_fmt yuv420p -threads 4 $FILE.mp4
```

where `$Resolution` $\in \{1920\text{x}1080, 640\text{x}360\}$, and `$CRF` $\in \{17, 20, 22, 27, 32, 37, 42, 47\}$.

## G  MORE EXPERIMENTAL RESULTS

### G.1  IMAGE RECONSTRUCTION

In Table 8, we compare the image reconstruction results under bilinear interpolation. We conclude that varying interpolation changes the values but unalts the order of all methods.

### G.2  IMAGE COMPRESSION

In Figure 12, we compare the rate-distortion (RD) curve in terms of MS-SSIM *vs.* bpp (bits per pixel) on Kodak. We see BSQ consistently improves MS-SSIM with increasing BPP while VQ-based AE shows diminishing gain at higher BPP.

### G.3  VIDEO COMPRESSION

We compare the compression result on MCL-JCV and UVG in Figure 8. On UVG 1080P, our model is comparable to H.264 while being worse than HEVC and VCT (Mentzer et al., 2022). Note that our

---

[1]https://ffmpeg.org/

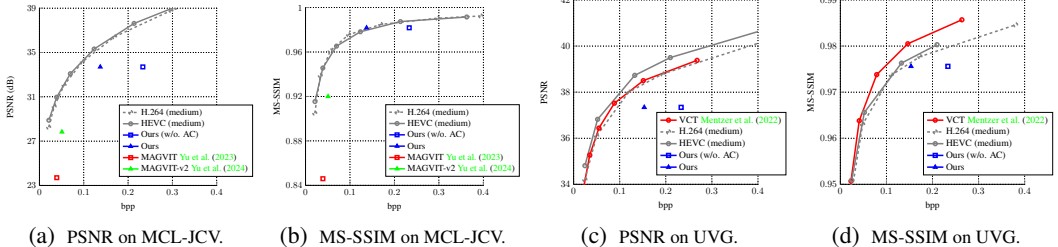

Figure 8: **Video compression results on MCL-JCV 640×360 and UVG 1920×1080.**

Table 9: **Comparisons of encoding/decoding speed.** [†]The number did not include the image encoder according to (Mentzer et al., 2022).

| Method | Resolution | Encode | EC | Decode | FPS |
|---|---|---|---|---|---|
| VCT (Mentzer et al., 2022) | 1920×1080 | [†]494 ms | 30.5 ms | 168 ms | 1.4 |
| H.264 | 1920×1080 | - | - | - | 2.6 |
| Ours | 1920×1080 | 55.8 ms | 42.2 ms | 64.8 ms | 6.1 |
| VCT (Mentzer et al., 2022) | 640×360 | [†]22.2 ms | 4.24 ms | 10.1 ms | 27.3 |
| H.264 | 640×360 | - | - | - | 22.4 |
| Ours | 640×360 | 6.2 ms | 4.69 ms | 7.2 ms | 55.2 |

model trains on UCF-101 which only has 9K 320×240 video clips encoded in MPEG-4 while VCT has been trained on a million high-resolution Internet video clips. We hypothesize that the gap will be mitigated by adding more diverse videos and removing compression artifacts from the training videos. Nevertheless, we show the potential advantage of our method in encoding and decoding speed in Table 9. Due to the simplicity of the Transformer-based encoder and decoder, our method runs faster than VCT.

# H  QUALITATIVE RESULTS

In Figure 9, we show reconstructed images produced by the proposed BSQ-ViT in comparison to the best prior work, SDXL-VAE (Podell et al., 2023). We can see that our method is able to preserve more details about high-frequency texture and fine-grained shape/geometry. BSQ-ViT often shows better reconstruction results for characters.

In Figure 10, we show sampled results produced by a Masked LM with the proposed BSQ-ViT in comparison to existing methods, BigGAN (Brock et al., 2018) and ADM (Dhariwal & Nichol, 2021). We also plot the samples from the ground-truth ILSVRC2012 validation set for reference. Our method produces competitive results with state-of-the-art methods.

In Figure 11, we show the compressed image produced by different methods under the same BPP used in Table 4a.

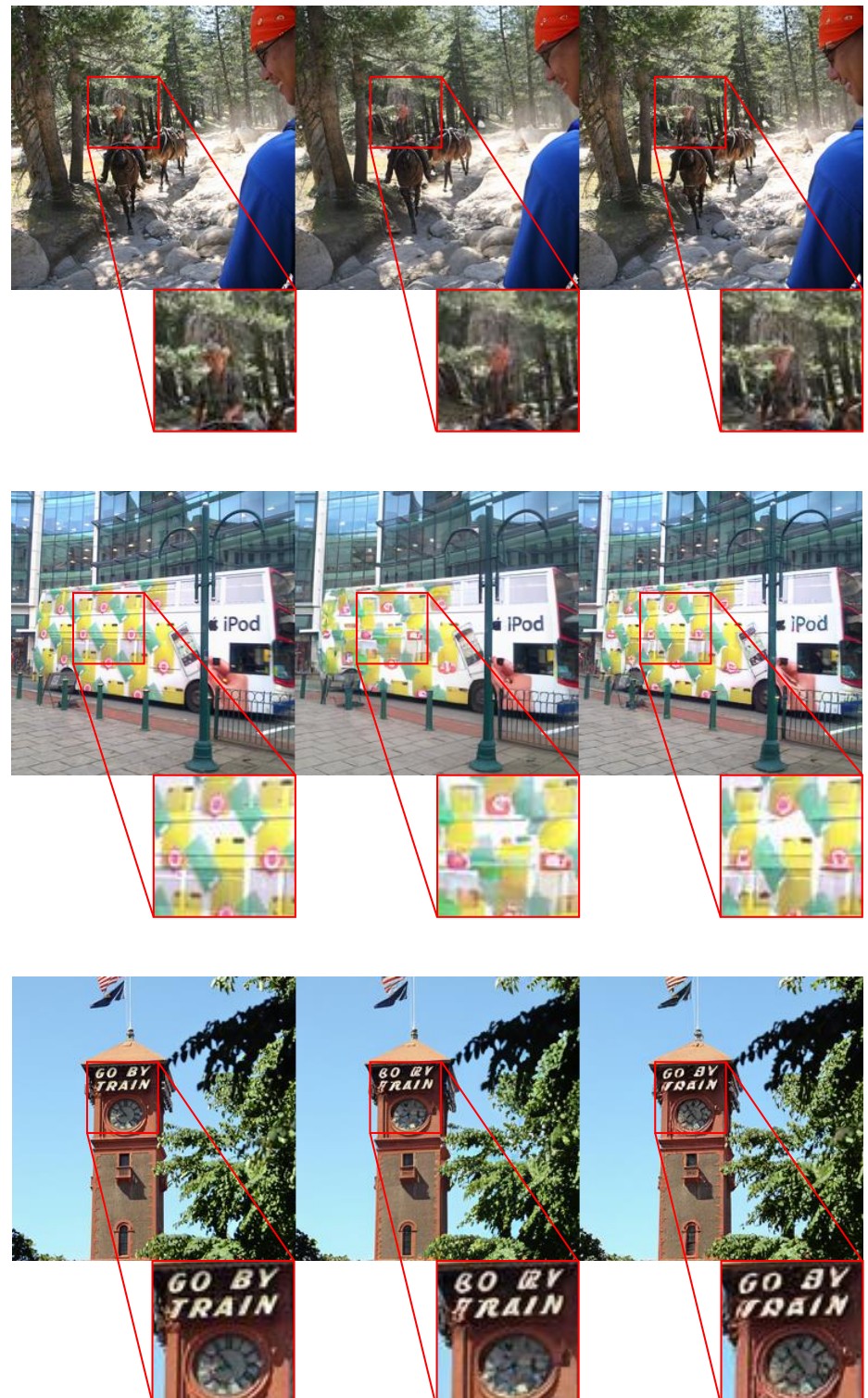

Figure 9: Reconstruction results of BSQ-ViT (**right**) compared to the original image (**left**) and SDXL-VAE (Podell et al., 2023) (**middle**). The three images are taken from COCO 2017val which are more scene-centric compared to ImageNet data that our model is trained on.

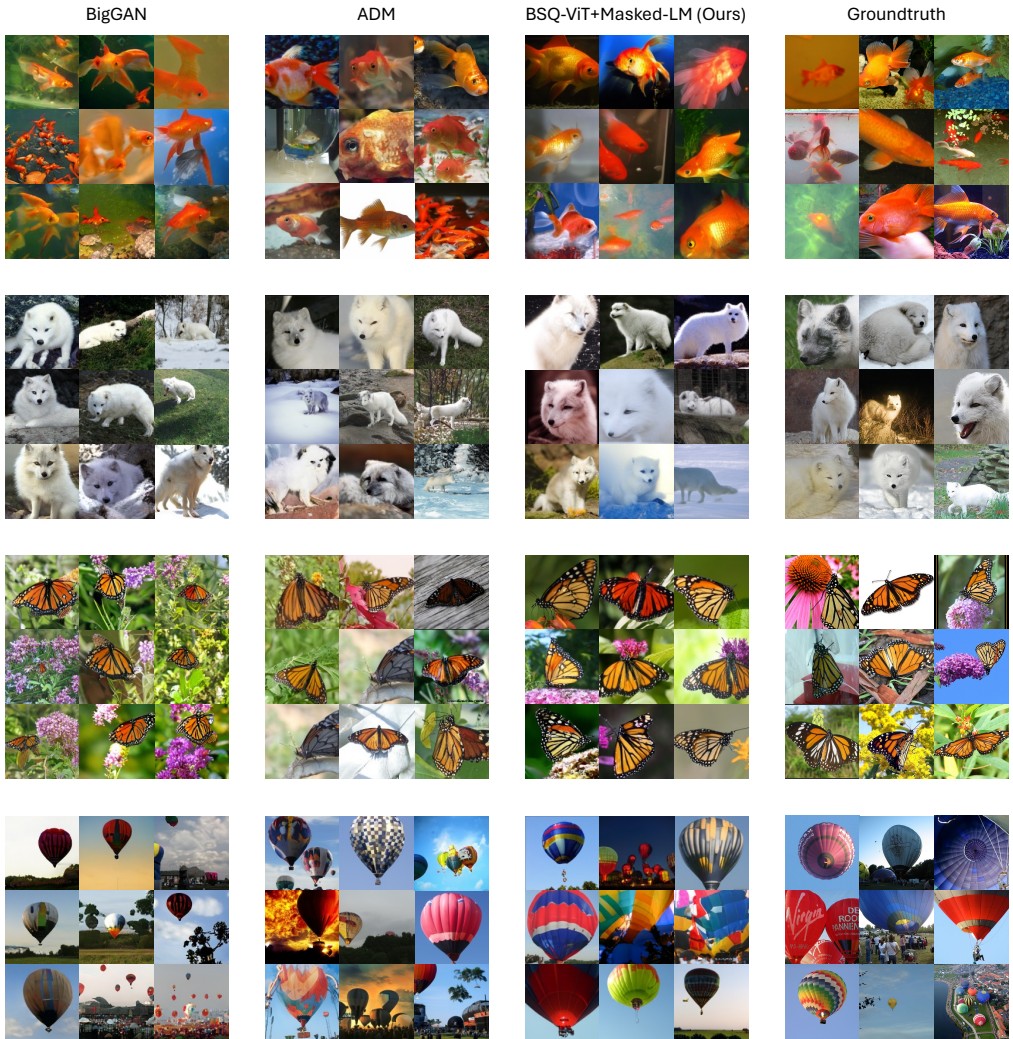

Figure 10: Sampled generation results of BSQ-ViT + Masked-LM (**second column from left**) compared to BigGAN (Brock et al., 2018) (**right**), ADM (Dhariwal & Nichol, 2021) (**second column from right**) and the original images (**left**). Classes are 1: goldfish, 279: arctic fox, 323: monarch butterfly, 417: balloon.

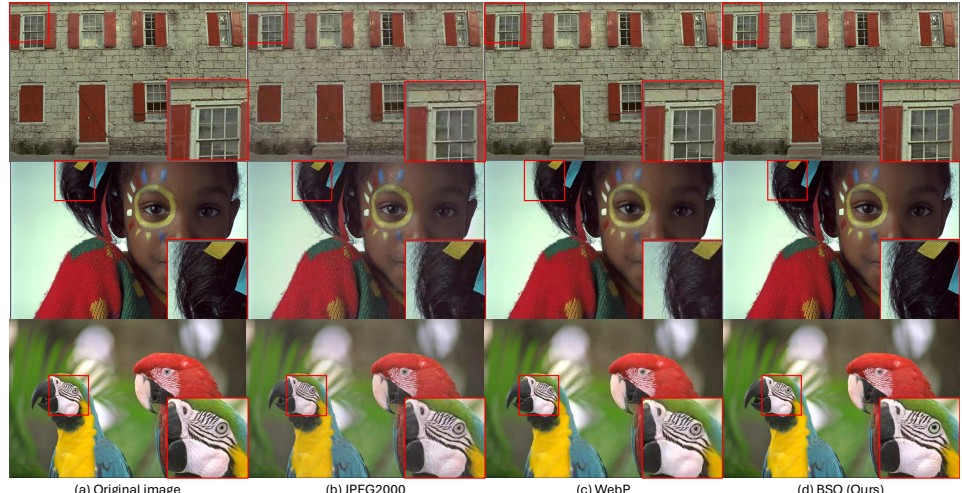

(a) Original image      (b) JPEG2000      (c) WebP      (d) BSQ (Ours)

Figure 11: Compression results of BSQ-ViT (**d**) compared to JPEG2000 (**b**), WebP (**c**) and the original images (**a**). The images are from Kodak (`kodim01`, `kodim15`, and `kodim23`).

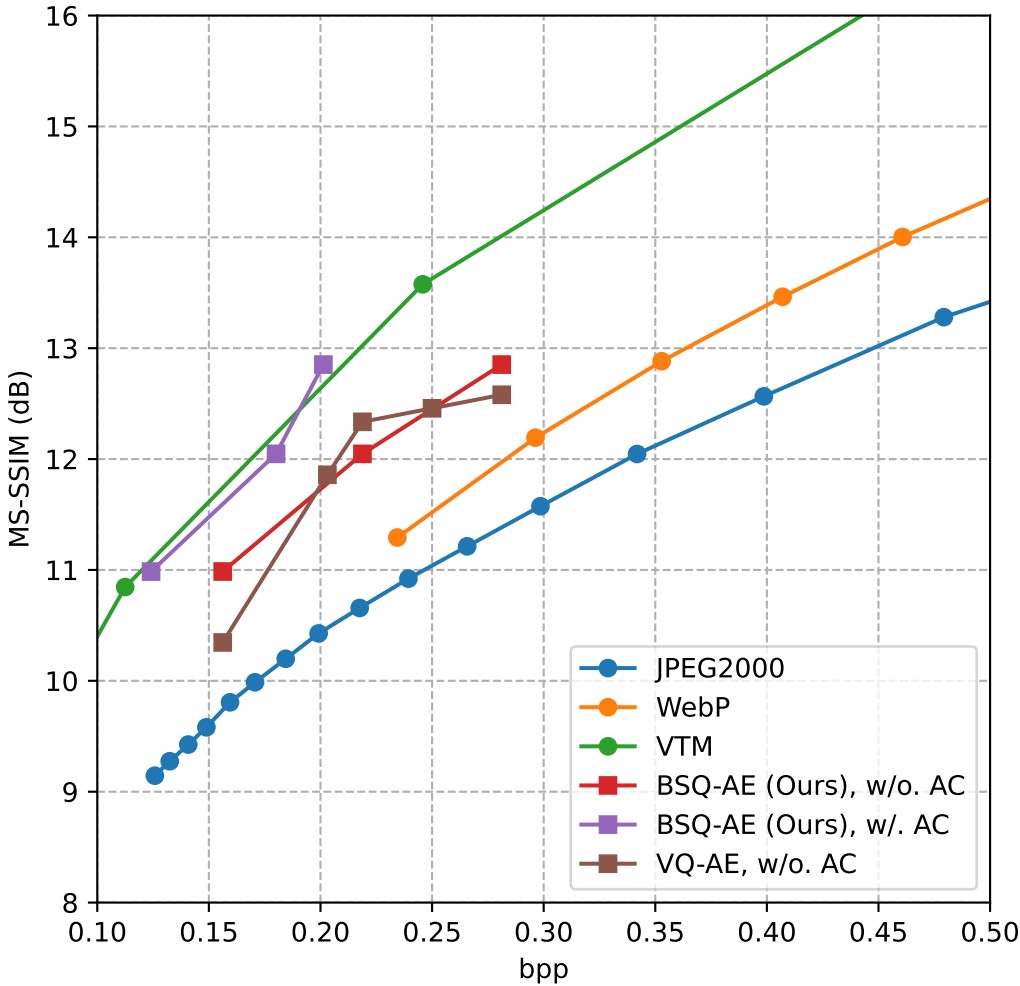

Figure 12: Image compression results on Kodak.

