# OpenReview forum: "Image and Video Tokenization with Binary Spherical Quantization"
_ICLR.cc/2025/Conference — ICLR 2025 Poster_

### Official Review · Reviewer_Aq9e · 2024-10-31

**Soundness:** 3
**Presentation:** 3
**Contribution:** 3
**Rating:** 6
**Confidence:** 4

**Summary:**

This paper proposes a visual tokenizer based on a Vision Transformer and Binary Spherical Quantization (BSQ).
The Transformer-based encoder-decoder leverages a block-wise causal mask and uses only visual tokens from the current or past timestamps for reconstruction.
BSQ first projects the high-dimensional visual embedding of the encoder to a lower-dimensional hypersphere and then applies binary quantization.
The transformer encoder, decoder, and BSQ are seamlessly integrated into the VQ-GAN framework and trained end-to-end.

The proposed visual tokenizer has several advantages in trading off  visual reconstruction quality and computational efficiency and supporting variable-length input. Experiments on visual reconstruction and compression are conducted to verify the performance.

**Strengths:**

Visual tokenizers are critical for visual modeling. The proposed Binary Spherical Quantization (BSQ) for unified image and video tokenization is novel and important.

Theoretical contribution is good. It is proved that the proposed BSQ has bounded quantization error.

The manuscript is well written. BSQ is well placed agains previous VQ and LFQ.

Extensive experiments on visual reconstruction, compression and generation show the better performance of BSQ.

**Weaknesses:**

My major concern is the fair comparison with VQ and LFQ. Though the ablation study is provided in  tab 5 with image reconstruction task on imagenet-1k, the resolution is only 128, and more tasks should be tested:

It seems to me the results on image is not as obvious as video. For example, in image reconstruction (tab 1), the proposed method has best image quality, but the parameter number is also larger. It is better to add Ours-VQ for better comparison, as done in tab 2 for video reconstruction.

In image compression, no VQ or LFQ based method is compared.

For image generation in tab 3, more steps are used for BSQ, what if we use the same steps as VQ and LFQ?

In L389, we get a direct comparison between VQ and BSQ, BSQ has more bits (18 vs 14) and the video quality is only comparable with VQ.





Minor
The derivation for eq. 13 is not given. Please provide a proof or give a reference here.
There exist some typos in the derivations.
In Line 878, the p shuld be Q.
In line 886, the p should be q^hat.

**Questions:**

My major concerns in the weakness part can be summarized as the direct comparison between BSQ, VQ and LFQ on representative image and video tasks are not adequate. As a general tokenizer for image and video, it is important to make direct comparisons with strictly controlled variables.

---

> ### Author Response · Authors · 2024-11-25
>
> ### 1. VQ in image reconstruction
>
> ~~We will follow up with the results once the training is done.~~
> We report our reproduced VQ with 18 bits on ImageNet 256x256 below (BSQ numbers are from Table 1)
> | Method | #bits | PSNR$\uparrow$ | SSIM$\uparrow$ | LPIPS$\downarrow$ | rFID$\downarrow$ |
> | --------- | ----- | ------- | ------ | ------- | ------ |
> |  VQ       |  18    |  25.07 | .7276  | .0717 | 1.68 |
> |  BSQ (Ours) | 18 | 25.36 | .7578 | .767 | 1.14 |
>
> Although the parameter number in ViT is larger than CNN-based approaches, the measured FLOPs are significantly smaller than CNNs. We compare SD-VAE 1.x and BSQ-ViT in the table and observe that ViT requires 36% FLOPs compared to CNN in SD-VAE.
>
> |      | #params | FLOPs | Throughput (images/sec) |
> | --  |  ---------  |  ------  | --- |
> | SD-VAE 1.x | 68M | 342 G | 22.4 |
> | BSQ-ViT (ours) | 174M | 125 G  | 45.1 |
>
> ### 2. VQ or LFQ-based method
>
> We add VQ and LFQ results below. For LFQ, we use the checkpoint from OpenMAGVIT2, an open-source implementation of MAGVIT2.
>
>
> | Method | BPP | PSNR$\uparrow$ | MS-SSIM$\uparrow$ | LPIPS$\downarrow$ |
> | --------- | ----- | --------------------| ------------------------- | ----------------------- |
> | JPEG2000 | 0.2986 | 29.192 | 11.574 | .1892 |
> | WebP        | 0.2963 | 29.151 | 12.193 | .1655 |
> | MAGVIT2  | 0.2812 | 23.467 | 8.103 | .1260 |
> | VQ  | 0.2812 | 26.987 | 12.580 | .0944 |
> | BSQ (Ours)  | 0.2812 | 27.785 | 12.852 | **.0823** |
>
> ### 3. Image generation with the same number of steps.
>
> | Method | steps | FID$\downarrow$ | IS$\uparrow$ | Prec$\uparrow$ | Rec$\uparrow$ |
> | -------- | ------- | --------------------- | --------------- | ------------------- | ------------------ |
> | VQ | 12 | 9.4 | - | - | - |
> | FSQ | 12 | 8.5 | - | - | - |
> | BSQ | 12 | 5.69 | 48.5 | 0.85 | 0.42 |
> | BSQ |  32 | 5.44 | 139.6 | 0.80 | 0.50 |
>
> We added the result with the same number of steps. We see a slight decrease in FID. The Inception Score decreases more significantly. The reason is that during masked language modeling, we divide each BSQ token into two sub-groups. This increases the effective sequence length, requiring a larger number of decoding steps to generate more details.
>
>
> ### 4. Proof of Eq 13
> The sketch of deriving $d_{max} (\mathbf{u}, \mathbf{\hat{u}}) $ is as follows. For an arbitrary vector $\mathbf{u}$ residing on the unit hypersphere ($|\mathbf{u}|_2 = 1$), its quantized counterpart $\mathbf{\hat{u}} = \frac{1}{\sqrt{L}} \mathrm{sign}(\mathbf{u})$ also resides on the same unit hypershpere.  We can write the quantization error as $d(\mathbf{u}, \mathbf{\hat{u}}) = \sqrt{2 - 2 \mathbf{u}\cdot\mathbf{\hat{u}}}$. The maximum of $d(\mathbf{u}, \mathbf{\hat{u}})$ is equivalent to finding the minimum of $\mathbf{u} \cdot \mathbf{\hat{u}} = \sum_i u_i \hat{u}_i$. Note that $\hat{u}_i = \frac{1}{\sqrt{L}} \mathrm{sign}(u_i) $. This leads to $u_i \hat{u}_i = \frac{1}{\sqrt{L}} | u_i | $ and $ \mathbf{u} \cdot \mathbf{\hat{u}} = \frac{| \mathbf{u} |_1} {\sqrt{L}}$. Minimizing $\mathbf{u} \cdot \mathbf{\hat{u}}$ is then equivalent to finding an $\ell_2$ unit vector within the minimal $\ell_1$ norm, which is known to be achieved only if the vector $\mathbf{u}$ is a one-hot vector and the minimal $\ell_1$ norm is 1.
>
> This leads to $d_{max} (\mathbf{u}, \mathbf{\hat{u}}) $ being $\sqrt{2 - \frac{2}{\sqrt{L}}}$ .
>
>
> We have added the proof in the appendix. Please refer to the rebuttal revision on Line 936.
>
> We thank the reviewer for spotting the typos. We have fixed all of them in the rebuttal revision, highlighted in red underlined text.

---

> > ### Comment · Reviewer_Aq9e · 2024-11-27
> >
> > Thanks for the reply. I am considering other reviewer comments carefully.

---

### Official Review · Reviewer_PNuG · 2024-11-01

**Soundness:** 3
**Presentation:** 3
**Contribution:** 3
**Rating:** 6
**Confidence:** 3

**Summary:**

The paper introduces a transformer-based image and video tokenizer that uses Binary Spherical Quantization (BSQ). By projecting high-dimensional visual embeddings onto a lower-dimensional hypersphere and applying binary quantization, BSQ achieves a bounded quantization error. The authors demonstrate that the proposed BSQ outperforms previous methods in image and video reconstruction, image generation, and compression tasks.

**Strengths:**

1. The idea of projecting high-dimensional visual embeddings onto a lower-dimensional hypersphere is straightforward yet effective.

2. The motivation is clear, and the overall presentation is coherent and easy to follow. The experiments are comprehensive and provide convincing evidence to support the approach.

3. The BSQ-ViT model achieves competitive performance in diverse tasks such as image/video reconstruction, generation, and compression.

**Weaknesses:**

1. This method uses a transformer encoder and decoder, which limit the flexibility in resolution. How do the authors address this issue?

2. For image and video compression results, it would be beneficial to include an LPIPS comparison to assess perceptual performance.

3. There are a few minor issues. 1) In Eq. (7), $\hat{q}(c|u) = \frac{\exp(\tau c^{T}u)}{\sum_{c \in C_{BSQ}} \exp(\tau c^{T}u)}$ might need to be revised to $\hat{q}(c|u) = \frac{\exp(2 \tau c^{T}u)}{\sum_{c \in C_{BSQ}} \exp(2 \tau c^{T}u)}$ ? 2) On page 16, line 837, $\frac{e^{\tau u_d \hat{c}_d}}{e^{\tau u_d \hat{c}_d} + e^{\tau u_d \hat{c}_d}}$ should be corrected to $\frac{e^{\tau u_d \hat{c}_d}}{e^{\tau u_d \hat{c}_d} + e^{-\tau u_d \hat{c}_d}}$. 3) On page 3, line 118, it would be clearer to use different symbols for the downsample factor $q$ and the bottleneck module $q$.

**Questions:**

1. For image compression, why not use arithmetic coding to improve compression performance? Is intra-frame information used for video compression? If not, why not?

---

> ### Author Response · Authors · 2024-11-25
>
> ### 1. How to address different resolutions.
>
> In our reconstruction experiments, the input for image/video reconstruction experiments is either 128x128 or 256x256. Therefore, we fix the image resolution to be 128 or 256 and train two models from scratch separately. For image/video compression experiments, we partition the image/video into multiple 128x128 crops and apply our BSQ-ViT(128x128) on all crops. For cases where the edge is not divisible by 128 (e.g. UVG-1080p), we pad the last row/column to be 128x128 with repeat padding.
>
> It is also possible to adapt the model with 128x128 inputs to accept 256x256 inputs by linearly interpolating the patch embedding layer and fine-tuning the model with the new input size, e.g. like FlexViT (https://arxiv.org/pdf/2212.08013).
>
> ### 2. LPIPS comparison.
>
> This is a great suggestion! We include LPIPS scores for image and video compression benchmarks.
>
> | Method | BPP | LPIPS$\downarrow$ |
> | --------- | ----- | ----------------------- |
> | JPEG2000 | 0.2986 | .1892 |
> | WebP        | 0.2963 | .1655 |
> | Ours          | 0.2812 | **.0823** |
>
> | Method | BPP | LPIPS$\downarrow$ |
> | --------- | ----- | ----------------------- |
> | MAGVIT    | 0.0391 | .144 |
> | MAGVIT-v2 | 0.0508 | .104 |
> | H.264 | 0.1373 | 0.949 |
> | H.265 | 0.1373 | 0.908 |
> | Ours (w/. AC)     | 0.1373 | **.0501** |
>
> We can see that our method outperforms JPEG2000 and WebP significantly. We also show three compressed images from the Kodak dataset at https://ibb.co/5GH1V1J. We see the compressed images generated by BSQ-ViT preserve more details. On the MCL-JCV (640x360) Video compression benchmark, our method also yields a much lower LPIPS score than H.264 and H.265 under the same BPP.
>
> ### 3. Minor issues.
>
> Thanks for spotting these issues! (1) should be correct. (2) and (3) are typos and we have fixed them in the rebuttal revision.
>
> ### 4. Arithmetic Coding for image compression.
> This is a great point! We train an auto-regressive transformer to model the conditional distribution for arithmetic coding. The average BPP reduces from 0.2812 to 0.2073, a 26% reduction (when $L=18$). This result is already close to the VTM result reported at https://github.com/InterDigitalInc/CompressAI/blob/v1.2.6/results/image/kodak/vtm.json. ~~We will update the number once the auto-regressive transformer finishes training~~. We update the rate-distortion curve in the Appendix Figure 12.
>
> ### 5. Intra-frame information in video compression.
>
> Video compression **uses** intra-frame information in two aspects. First, the Transformer-based autoencoder with blockwise casual mask models intra-frame information via self-attention within each block. The block causal masks ensure that each token can attend to all other tokens in the same frame as well as tokens in all previous frames. Second, the auto-regressive transformer for arithmetic coding jointly models the intra-frame and inter-frame information.

---

### Official Review · Reviewer_zX4y · 2024-11-02

**Soundness:** 2
**Presentation:** 2
**Contribution:** 2
**Rating:** 6
**Confidence:** 5

**Summary:**

This paper propose a transformer-based image and video tokenizer with Binary Spherical Quantization, which projects the high-dimensional embedding to lower-dimensional hypersphere and applies binary quantization. BSQ is parameter-efficient without an explicit codebook, scalable to token dimensions and compact. The experiments show that the proposed method achieves comparable compression performance with JPEG2000/WebP for images and H.264/H.265 for videos. And it enables masked language models to achieve competitive image synthesis quality to GAN and diffusion methods.

**Strengths:**

The Binary Spherical Quantization seems to show more effective training of the qunatization bottleneck. Analysis shows that the proposed method can provide fast speed and good performance.

**Weaknesses:**

- Lack of comparison at different bitrate range for visual compression results. Table 4 only provides BPP, PSNR and MS-SSIM for one bitrate point. However, visual compression tasks usually require showing a Rate-Distortion curves and compare at different bitrate points. Your can use BD-Rate metric for more reasonable comparison and analyze the results at low bitrate and high bitrate.

- Test settings for ablation study. Please provide more experiment setting details. In Table 5, do VQ, LFQ and BSQ use the same tokenizers? I observed that the metrics vary significantly across different quantization methods. To alleviate the influence of training and focus on the quantization bottleneck, it is better to: 1. use the same tokenizer 2. freeze the other network parts and only train the information bottleneck parts.

- Complexity. It is better to provide the computation complexity and encoding/decoding time for comparison.

**Questions:**

See the weakness part

---

> ### Author Response · Authors · 2024-11-25
>
> ### 1. Rate-Distortion curve.
>
> This is a great suggestion! We provide a more detailed rate-distortion (RD) comparison at https://ibb.co/GpBV47b. The figure can also be found in the Appendix Figure 12. From the RD curve, we can see that BSQ consistently improves MS-SSIM with increasing bpp. VQ-AE shows diminishing gain at higher bpp. We ascribe this to the difficulty of training VQ when the codebook size increases. As per Reviewer PNuG's comments, we run an auto-regressive Transformer to compress the image tokens with arithmetic coding. This further leads to a ~26% reduction of bpp and the final result is very close to the VTM result reported at https://github.com/InterDigitalInc/CompressAI/blob/v1.2.6/results/image/kodak/vtm.json. We will add more data points in the final revision.
>
> ### 2. Testing settings for ablation studies.
>
> (1) We use the same encoder-decoder while only changing the quantization bottleneck. We train all models on ImageNet-1k 128x128 from scratch.
>
> (2) An alternative way is to train an AE first and learn the quantizer on top of the encoder while freezing the encoder and decoder. ~~We will follow up with the result once the training is done.~~
> We observe separating training encoder-decoder and bottlenecks does not yield satisfying results for all quantization methods (reconstruction loss shown in the table below).
>
> | Method | $\ell_{recon}$ |
> | --------- | ---------------- |
> | BSQ (e2e) |     0.0977    |
> | VQ         |      0.1922      |
> | BSQ      |      0.2401      |
> | LFQ       |      0.3177      |
>
> The implicit codebook in BSQ and LFQ employs geometric prior of the embedding at the bottleneck, whereas training an Auto-Encoder does not. On the contrary, the explicit codebook in VQ is freely learnable and has more parameters ($O(2^L\times d)$ vs. $O(L\times d)$ in BSQ and LFQ), leading to slightly better performance. Nevertheless, none of the methods are comparable to the default end-to-end training setup ("e2e" in the table). It is hard for an autoencoder (without a quantization bottleneck) to learn low-variance latent space without adding inductive biases [1,2]. Therefore, we believe it makes more sense to train an auto-encoder with a quantization bottleneck in an end-to-end manner which is the default setup in our ablation.
>
> [1] Kingma and Welling. "Auto-encoding variational bayes." ICLR 2014.
> [2] Higgins, et al. "beta-vae: Learning basic visual concepts with a constrained variational framework." ICLR 2017.
>
>
> ### 3. Computation complexity and encoding/decoding time for comparison
>
> We left the comparison of computation costs in the appendix due to limited space. Please refer to Table 9. Due to the simplicity of the Transformer-based encoder and decoder, our method runs faster than VCT. For H.264, we use FFmpeg’s API and only report the overall FPS.

---

### Official Review · Reviewer_GS7J · 2024-11-02

**Soundness:** 3
**Presentation:** 3
**Contribution:** 2
**Rating:** 6
**Confidence:** 4

**Summary:**

This paper proposes a novel image and video tokenizer, BSQ-ViT, based on Binary Spherical Quantization (BSQ) integrated with a transformer architecture. The proposed method achieves state-of-the-art performance in image and video reconstruction with significant improvements in computational efficiency. It introduces a block-wise causal masking for video tokenization, supports variable-length videos, and delivers competitive results in visual compression against standards like JPEG2000 and H.264. BSQ-ViT also shows promising results in image generation.

**Strengths:**

1. The paper presents an innovative quantization method (BSQ) that addresses the limitations of existing vector quantization approaches by offering a more efficient and scalable solution.
2. Extensive experiments on benchmarks such as ImageNet and UCF-101 demonstrate that BSQ-ViT significantly improves reconstruction quality, outperforming prior methods in terms of speed and fidelity.
3. The methodology is clearly explained with detailed comparisons to related work, and the theoretical basis of BSQ is well-supported with mathematical derivations and experimental validation.
4. BSQ-ViT's ability to handle both image and video tokenization and perform well on diverse tasks such as compression and generation showcases its generalizability.

**Weaknesses:**

1. While the transformer architecture is explored, the paper does not demonstrate the effectiveness of  BSQ within a CNN-based model.
2. The paper provides limited comparative data in video reconstruction, reducing the robustness of the comparison. Additionally, while block-wise causal attention is noted to impact performance, the study lacks experiments on BSQ without this causal masking.
3. The reported image and video compression results are better on MS-SSIM, potentially  due to the inclusion of perceptual losses like LPIPS and GAN loss in the loss function. However, the lack of subjective evaluation metrics limits the strength of the comparison.
4. When BSQ and VQ use the same number of indexes, BSQ performs less effectively, and codebook utilization decreases as the bits increase.
5. In the ablation experiments, the reproduction of the LFQ, which serves as the basis for this work, is too poor in terms of behavior and code usage. And it is premature to conclude that LFQ is less compatible with transformer-based codecs.

**Questions:**

see the weaknesses.

---

> ### Author Response · Authors · 2024-11-25
>
> ### 1. Effectiveness of BSQ within a CNN-based model.
>
> BSQ is effective within a CNN-based autoencoder. We use the CNN-based encoder-decoder similar to SD-VAE. The only architectural change is that the encoder outputs an $L$-dim vector which is then $\ell_2$-normalized and fed into binary quantization without an external projection layer ($\mathrm{z} \rightarrow \mathrm{v} $) in ViT. ~~We will post the final result once the training is done.~~ Update: We train a BSQ-ViT and BSQ-CNN with a shorter schedule (200K iterations). The results are shown below. We can see that BSQ-CNN works comparably well with BSQ-ViT, validating the effectiveness of BSQ within a CNN-based model.
>
> | backbone | PSNR$\uparrow$ | SSIM$\uparrow$ | LPIPS$\downarrow$ | FID$\downarrow$ |
> | ----------- | ----------- | ------- | ------- | ---- |
> |   CNN       |   25.14 | .7680 | .0827 | 7.06 |
> |   ViT         |   25.01 | .7700 | .0800 | 6.46 |
>
> ### 2. BSQ without this causal masking.
>
> We show VQ without blockwise causal masking in the last but fourth line in Table 2. We denote it with “non-BC” ViT for short.
>
> | Method | backbone | PSNR$\uparrow$ | SSIM$\uparrow$ | LPIPS$\downarrow$ | rFVD$\downarrow$ |
> |  -------- | ----------- | -------------------- | -------------------- | ------------------------ | ---------------------- |
> | Ours   | non-BC ViT |     33.06               |    .9518                 |         0.0223                |      9.16                   |
> | Ours   |  BC ViT       |     32.81               |     .9496                |          0.0236               |      10.76                  |
>
> We will add BSQ without blockwise causal masking in the final revision.
>
> ### 3. Subjective evaluation metrics of image compression.
>
> We show three compressed images from the Kodak dataset at https://ibb.co/5GH1V1J. The figure has also been added in the appendix (Figure 11)
>
> ### 4. BSQ vs VQ.
>
> In the **ideal** scenario, VQ works better than BSQ in that it leverages the entire high-dimensional space while BSQ only leverages the hypercube corners. However, **in reality**, VQ is difficult to train when the codebook size increases, as widely acknowledged in previous literature and shown in our ablation (Table 5).  We also provide a more detailed rate-distortion (RD) comparison at https://ibb.co/GpBV47b. The figure can also be found in the Appendix Figure 12. We can see VQ-AE falls behind BSQ-AE when bpp increases. In addition, given the same bottleneck of $L$ bits, VQ's parameter space is $O(d\times 2^L)$ while  BSQ is only $O(d\times L)$.
>
> ### 5. Reproducing LFQ.
>
> When we submitted the manuscript we re-implemented the LFQ following the parameters in the original paper. To ensure the results are reproducible, in the discussion period, we additionally implemented LFQ using OpenMAGVIT2's implementation (https://github.com/TencentARC/Open-MAGVIT2). We use their recommended hyper-parameters. Since the quantization loss uses a different set of weights, we compare the reconstruction loss at https://ibb.co/PQNHWtv. We can see the reconstruction loss of LFQ converges much slower. This is consistent with our original ablation in the submission that LFQ is difficult to train within the ViT-based autoencoder architecture *due to unbounded quantization error*. ~~We will report the final revision.~~ Update: We report the best result we achieved during the discussion period compared to the one we got when we submitted the manuscript. PSNR, SSIM, and LPIPS score improve while rFID remains similar. Neither case is comparable to the proposed BSQ.
>
> | Method                        | PSNR$\uparrow$ | SSIM$\uparrow$ | LPIPS$\downarrow$ | rFID$\downarrow$ |
> | ------------------------- | ------- | ------ | ------- | ------- |
> | LFQ reproduced by us | 18.58 | .4828 | .2951 | 30.7 |
> | LFQ reproduced by OpenMAGVIT2 | 21.68 | .6332 | .1939 |  31.6 |
> | BSQ                            | 25.97 | .7990 | .0629 | 2.66 |

---

> > ### Comment · Reviewer_GS7J · 2024-11-26
> > **response**
> >
> > I appreciate the authors' rebuttal and the additional experiments provided. However, I still have a few concerns:
> >
> > 1. While you suggest that LFQ is challenging to train on ViT (which might be) and you provide results for BSQ+CNN, you should also include results for LFQ+CNN for fair comparison.
> > 2. Thank you for providing visual examples. However, I remain uncertain about BSQ's performance on video reconstruction and compression, particularly regarding video continuity and related aspects. Additionally, it is essential to calculate subjective quality metrics, such as LPIPS and VMAF, for image and video compression tasks to offer a more comprehensive evaluation of the method's performance.

---

> ### Author Response · Authors · 2024-11-29
>
> Thank you for the follow-up comments!
>
> 1. We include the result of LFQ+CNN for fair comparison below. With the same CNN backbone, BSQ achieves better PSNR, SSIM, LPIPS, and comparable FID scores.
>
> | backbone | Quantizer | PSNR$\uparrow$ | SSIM$\uparrow$ | LPIPS$\downarrow$ | FID$\downarrow$ |
> | ----------- | ----------- | ----------- | ------- | ------- | ---- |
> |   CNN       | LFQ |   24.26 | .7095 | .1353 | 6.73 |
> |   CNN       | BSQ |   25.14 | .7680 | .0827 | 7.06 |
> |   ViT         | LFQ   | 21.68 | .6332 |  .1939 | 31.6 |
> |   ViT         | BSQ  | 25.01 | .7700 | .0800 | 6.46 |
>
>
> 2. We have added the LPIPS scores in the revised manuscript as requested by Reviewer PNuG. We further compute the VMAF (v0.6.1) metric. We post both numbers below.
>
> | Method |  BPP     | LPIPS$\downarrow$ | VMAF$\uparrow$ |
> | --------- | -------- | ------- |  ----- |
> | H.264    | 0.1373 | .0949 | 92.61 |
> | H.265    | 0.1373 | .0908 | 92.42 |
> |  Ours (w/.AC) | 0.1373 | .0501 |  86.46 |
>
> Our method outperforms H.264 and H.265 in terms of LPIPS but underperforms in terms of VMAF. We look at the results and observe that our method works better on videos with slow motion but falls behind on videos with large motions. We uploaded two representative videos at https://file.io/c7WKp9Gacb45. The reason is probably because we fine-tune the video auto-encoder on UCF-101 only, whose diversity and quality limit the performance.

---

### Meta-Review · Area_Chair_FLCs · 2024-12-17

**Metareview:**

The authors propose a novel image and video tokenizer based on a binary alphabet. It addresses difficulties training tokenizers based on larger alphabets, and shows solid performance and speed improvements while being scalable. The authors satisfactorily address concerns  regarding fair empirical evaluation.

Given the wide usage of VQ-like tokenizers in the larger ICLR community, I believe it may be of value to accept this paper as a spotlight, in order to make the existence of competitive alternatives more widely known. However, in the presence of time or space constraints, a poster would be appropriate as well, given the reviewer's ratings.

**Additional Comments On Reviewer Discussion:**

The reviewers mostly asked for additional empirical evaluations. While there was little feedback from them regarding the authors' rebuttal, I considered the rebuttal and think it sufficiently addresses the concerns raised by the reviewers.

---

### Decision · Program_Chairs · 2025-01-22

Accept (Poster)